# CD301b$^+$ dendritic cell-derived IL-2 dictates CD4$^+$ T helper cell differentiation

Naoya Tatsumi [1,2], Jihad El-Fenej[1,2], Alejandro Davila-Pagan[1,2] & Yosuke Kumamoto [1,2] ✉

T helper (Th) cell differentiation is fundamental to functional adaptive immunity. Different subsets of dendritic cells (DC) preferentially induce different types of Th cells, but the DC-derived mechanism for Th type 2 (Th2) differentiation is not fully understood. Here, we show that in mice, CD301b$^+$ DCs, a major Th2-inducing DC subset, drive Th2 differentiation through cognate interaction by rapidly inducing IL-2 receptor signalling in CD4$^+$ T cells. Mechanistically, CD40 engagement prompts IL-2 production selectively from CD301b$^+$ DCs to maximize CD25 expression in CD4$^+$ T cells, which instructs the Th2 fate decision, while simultaneously skewing CD4$^+$ T cells away from the T follicular helper fate. Moreover, CD301b$^+$ DCs utilize their own CD25 to facilitate directed action of IL-2 toward cognate CD4$^+$ T cells, as genetic deletion of CD25 in CD301b$^+$ DCs results in reduced IL-2-mediated signalling in antigen-specific CD4$^+$ T cells and hence their Th2 differentiation. These results highlight the critical role of DC-intrinsic CD40−IL-2 axis in Th cell fate decision.

Activation of antigen-specific CD4$^+$ T cells by dendritic cells (DCs) is critical for initiating adaptive immune responses. Upon recognition of cognate peptide-major histocompatibility complex class II (MHCII) complex through the T cell receptor (TCR), naive CD4$^+$ T cells rapidly proliferate as they differentiate into distinct subsets of effector T helper (Th) cells such as Th1, Th2, and Th17 cells, each of which have unique functions in immunity[1]. DCs play a crucial role in both expansion and differentiation of antigen-specific CD4$^+$ T cells not only by presenting antigens with costimulation but also by producing cytokines that affect the Th cell fate decision[2]. Notably, different DC subsets have been shown to produce distinct patterns of cytokines upon stimulation and thereby preferentially induce different types of Th cells. For instance, among the three major DC subsets in mouse skin, including the dermal type 1 conventional DCs (cDC1s), the dermal cDC2s, and the epidermal Langerhans cells (LCs), which have a monocytic origin but have DC-like function, the dermal cDC1s preferentially induce Th1 cell differentiation by producing IL-12[3,4], whereas LCs induce Th17 cell differentiation through IL-6 production[3,5]. Such division of labor between DC subsets is not unique to the skin, though the capacity to induce specific types of Th cells is not always confined to the same DC subset, as DC subsets other than cDC1s can be a critical source of IL-12 for inducing Th1 cell differentiation when exposed to certain stimuli[6,7]. Likewise, in organs outside of the skin where LCs do not exist, the Th17 cell differentiation is induced by a specialized subset of cDC2s that produce high levels of IL-23 and/or IL-6[8–11]. In contrast, the dermal cDC2s and cDC2s elsewhere have been shown to be required for the differentiation of Th2 cells[12–16], which play a crucial role in the development of allergic diseases and protection against helminth parasites. However, unlike the Th1 and Th17 fate induced by defined cytokines, the DC-derived factors acting directly on CD4$^+$ T cells for instructing the Th2 cell fate have not been clearly identified.

We and others have shown previously that CD301b$^+$ DCs, a major, migratory subset of cDC2s, are required specifically for the Th2 cell differentiation of antigen-specific CD4$^+$ T cells induced by allergens, adjuvants, or by helminth infection[12,13,17,18]. Diphtheria toxin (DT)-induced depletion of CD301b$^+$ DCs in mice expressing the DT receptor (DTR) under the regulation of the *Mgl2* (encoding CD301b) promoter (Mgl2-DTR mice) resulted not only in delayed priming but also in reduced Th2 cell differentiation with a compensatory increase in Th1, Th17 and T follicular helper (Tfh) cell differentiation[19,20]. These findings suggest that CD301b$^+$ DCs dictate the bifurcation of effector CD4$^+$ T cell fate, but its mechanism remains unknown.

[1]Center for Immunity and Inflammation, Rutgers New Jersey Medical School, Newark, NJ, USA. [2]Department of Pathology, Immunology and Laboratory Medicine, Rutgers New Jersey Medical School, Newark, NJ, USA. ✉e-mail: yosuke.kumamoto@rutgers.edu

Here, we show that the Th2 cell fate instruction by CD301b[+] DCs requires MHCII- and CD40-dependent cognate interaction with CD4[+] T cells. Mechanistically, the CD40 ligation induces IL-2 production specifically from CD301b[+] DCs, which is required for the maximal CD25 upregulation and the downstream IL-2 receptor (IL-2R) signaling in antigen-specific CD4[+] T cells. The full expression of CD25 in antigen-specific CD4[+] T cells is more strictly required for their differentiation into Th2 than for Th1 cells. In addition, the IL-2R signaling in the cognate CD4[+] T cells early after priming requires CD25 expression in CD301b[+] DCs, suggesting that the DC-derived CD25 facilitates the directed action of IL-2. Furthermore, the CD301b[+] DC-intrinsic CD40–IL-2 axis skews CD4[+] T cells away from differentiation into Tfh cells. These data highlight the critical role of DC-intrinsic CD40–IL-2 axis in the bifurcation of Th cell fate.

## Results

### Direct antigen presentation by CD301b[+] DCs is required for Th2 cell differentiation

CD301b[+] DCs are required for Th2 cell differentiation induced by papain (protease allergen), alum (type 2 adjuvant), and *Nippostrongylus brasiliensis* (*Nb*) (helminth parasite)[13], but whether they instruct the Th2 cell fate through a cognate interaction or via antigen-independent cues (e.g., cytokines) remains unclear. To address this question, we examined Th2 cell differentiation in mice lacking MHCII specifically in CD301b[+] DCs (*Mgl2[+/Cre];H2ab1[fl/fl]* mice, hereafter CD301b[ΔMHCII] mice)[20] by adoptively transferring carboxyfluorescein succinimidyl ester (CFSE)-labeled ovalbumin (OVA)-specific TCR transgenic CD4[+] T (OT-II) cells[21] and subcutaneously immunizing in the footpad with OVA plus papain (Fig. 1a). As reported previously, the Cre expression in the *Mgl2[Cre]* strain, examined by the inducible tdTomato (iTom) reporter (*Mgl2[+/Cre];R26[LSL-iTom]*) mice, was restricted to CD301b[+] PD-L2[+] migratory (MHCII[hi]) cDC2 (migDC2) and some CD301b[−] PD-L2[+] migDC2s, the latter of which presumably expressed a low level of CD301b that was not detectable due to the reduced CD301b protein expression from the *Mgl2[Cre]* allele[20], while the Cre expression was nearly absent from all other cell types including PD-L2[−] migDC2s, LCs, XCR1[+] migDC1s, and LN-resident cDCs (Supplementary Fig. 1a, b). Seven days after the immunization, the OT-II cells in the draining lymph node (dLN) of the CD301b[ΔMHCII] mice showed normal cell divisions with a mild increase in number (Fig. 1b, c), indicating that they were primed by antigen-presenting cells other than CD301b[+] DCs. However, as in CD301b[+] DC-depleted mice[13], the OT-II cells in CD301b[ΔMHCII] mice failed to develop IL-4-producing Th2 cells, while their differentiation into IFNγ-producing Th1 cells remained unchanged (Fig. 1d, e). Likewise, when mice were immunized with OVA plus alum, the Th2 differentiation of OT-II cells was impaired in CD301b[ΔMHCII] mice without affecting cell divisions, numbers, or differentiation into Th1 cells (Fig. 1f–h). In contrast, the expansion and Th1 differentiation of OT-II cells were not affected in the CD301b[ΔMHCII] mice when immunized with OVA plus a Th1-plarizing adjuvant CpG oligodeoxynucleotide (Fig. 1i–k). Lastly, as in CD301b[+] DC-depleted mice[13], the CD301b[ΔMHCII] mice had similar numbers of CD44[+] activated CD4[+] T cells but showed impaired IL-4 production from the activated CD4[+] T cells in the skin- and lung-dLNs upon subcutaneous infection with *Nb*, a model in which the parasites quickly migrate from the skin to the lung within one day, indicating the requirement of cognate interaction with CD301b[+] DC for the parasite-induced Th2 cell differentiation in the endogenous polyclonal CD4[+] T cells (Fig. 1l–n). Of note, the expression of CD301b is not a simple indicator for maturation status but is rather marking a specific cDC2 subset, as the Cre[+] and Cre[−] migDC2s in the *Mgl2[+/Cre];R26[LSL-iTom]* mice and CD301b[+] and CD301b[−] migDC2s in wild-type (WT) mice expressed CD86 at a comparable level (Supplementary Fig. 1c–g). These results indicate that cognate interaction with CD301b[+] DC is required specifically for the differentiation of Th2 cells.

### CD301b[+] DCs are required for inducing full CD25 expression in CD4[+] T cells

DCs send critical activation signals to antigen-specific naive CD4[+] T cells, leading to cell cycle entry and rapid upregulation of early activation markers such as CD69 and CD25 in T cells. We have shown previously that the depletion of CD301b[+] DCs results in an 8- to 16-hour delay in CD69 upregulation and cell cycle entry of antigen-specific CD4[+] T cells[20], but the link between the delayed activation and the selective impairment in Th2 differentiation remains unclear. To gain further insight into the potential defect in CD4[+] T cell activation in these mice, we transferred CFSE-labeled OT-II cells into DT-treated WT (CD301b[+] DC-intact) or Mgl2-DTR (CD301b[+] DC-depleted) mice 24 h after immunization with OVA plus papain in the footpad and then blocked further LN entry of T cells with anti-CD62L monoclonal antibody (mAb) 2 h later (Fig. 2a), which allows us to monitor the activation kinetics of OT-II cells in a synchronized manner[20]. In agreement with our previous report[20], the upregulation of CD69 in OT-II cells was delayed in CD301b[+] DC-depleted mice but was quickly recovered presumably due to antigen presentation by CD301b[−] DCs (Fig. 2b). In contrast, CD25 expression was induced in a subset of CD69[+] OT-II cells at a similar timing and peaked at 16 h after the dLN entry in both CD301b[+] DC-intact and CD301b[+] DC-depleted mice, but with significantly lower expression levels in the latter (Fig. 2b, c). The requirement of CD301b[+] DCs for the full CD25 upregulation is subset-specific, as the depletion of CD207[+] DCs, including LCs and dermal CD103[+] cDC1s[22–24], in CD207-DTR mice[25] did not affect CD25 expression (Fig. 2d).

CD25 (IL-2Rα) is the high-affinity subunit of the IL-2R complex. Intracellular staining of phosphorylated STAT5 (pSTAT5) in OT-II cells expressing the Nur77-GFP reporter (Nur77-GFP;OT-II), whose expression correlates with the TCR signal strength[26], showed a reduction of pSTAT5, the major signaling component of IL-2R, in OT-II cells primed in CD301b[+] DC-depleted mice (Fig. 2e, f). In contrast, like CD69, expression of the Nur77-GFP reporter soon after the dLN entry was lower in Mgl2-DTR than in WT mice (Fig. 2e, 0 h), but the comparable expression at a later time-point indicated a similar amount of TCR stimulation received by OT-II cells over time regardless of the CD301b[+] DC depletion status (Fig. 2e, 16 h). The pSTAT5 levels were also reduced in CD301b[ΔMHCII] mice (Fig. 2f), indicating the requirement of cognate interaction with CD301b[+] DC for optimal activation of STAT5 in OT-II cells. Furthermore, the subset-specific requirement of CD301b[+] DCs for the full CD25 upregulation and STAT5 phosphorylation in OT-II cells was also observed when the mice were immunized under a non-Th2 condition with Freund's complete adjuvant (FCA) (Fig. 2g–i). CD301b[+] DCs, but not CD207[+] DCs, were also required for the full upregulation of CD25 in OT-II cells in a more potent Th1 condition with CpG that did not induce any Th2 cells (Supplementary Fig. 2a–c), though they were dispensable for the Th1 cell differentiation (Fig. 1k). The expression of GATA-3, the master regulator of Th2 cell differentiation, was induced in a subset of Nur77-GFP[+] OT-II cells and showed positive correlation with pSTAT5 levels in the dLN of WT mice immunized with papain (Fig. 2j). The depletion of CD301b[+] DCs resulted in a reduction of GATA-3 in OT-II cells in association with the impaired activation of STAT5 (Fig. 2j, k). While OT-II cells were transferred into pre-immunized mice in the above data, similar results were obtained when the dLN was pre-seeded with naïve OT-II cells or harvested at later time-points (Supplementary Fig. 2d–h), indicating that the impaired Th2 differentiation in the absence of CD301b[+] DC-mediated priming was not due to a simple delay in IL-4 production. Collectively, these data indicate that CD301b[+] DCs deliver a qualitatively distinct signal required for the optimal CD25 upregulation and Th2 cell differentiation in antigen-specific CD4[+] T cells.

### Th2 cell differentiation relies more stringently on IL-2R signaling than Th1 cells

IL-2 plays fundamental roles in CD4[+] T cell biology[27,28]. While some studies have suggested the requirement of IL-2R signaling for Th2 cell

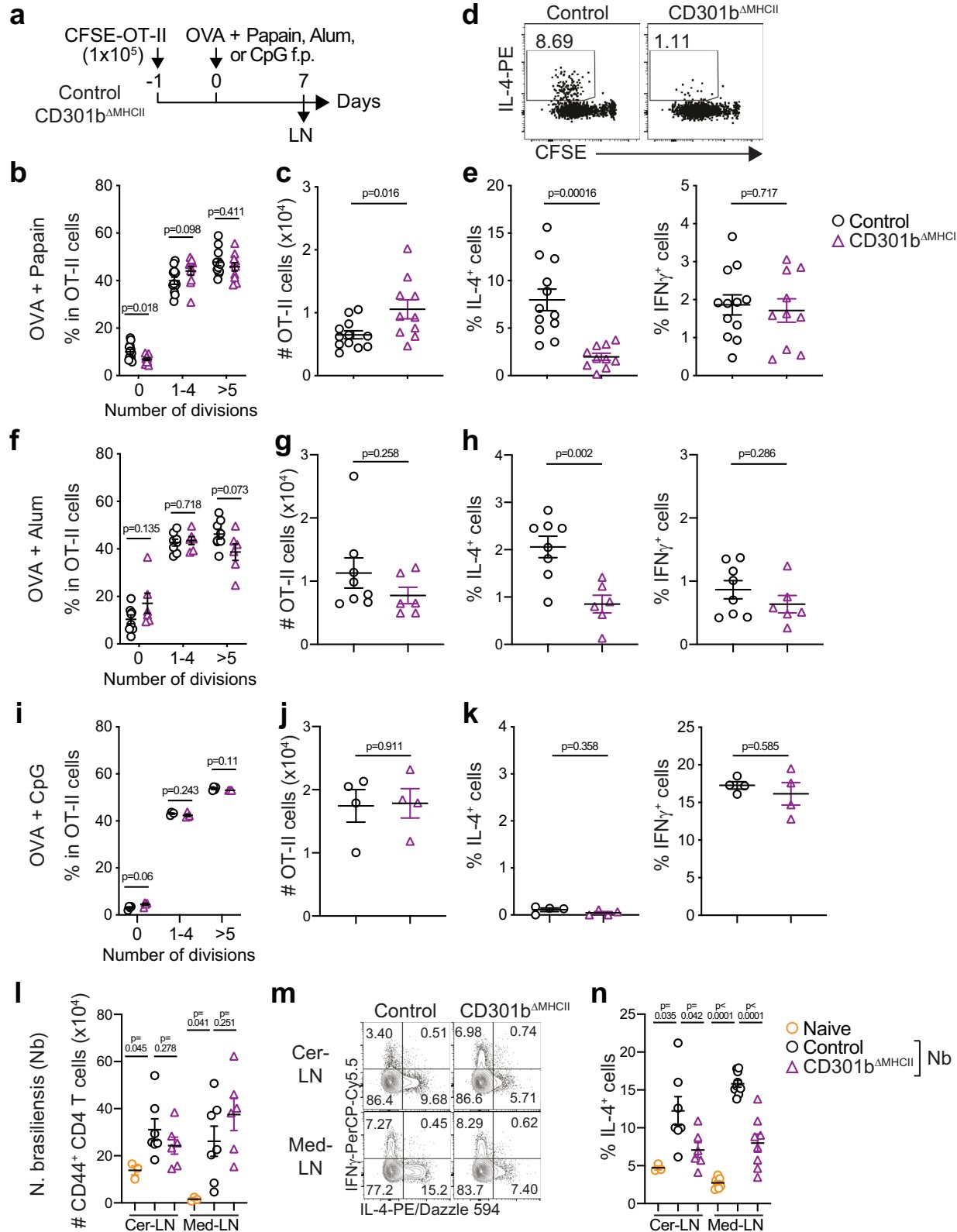

differentiation in vitro[29,30], its role in Th2-specific fate decision in vivo remains elusive, as abrogating IL-2R signaling often results in a broader defect in proliferation, differentiation, and/or survival of CD4+ T cells[31,32]. For instance, upon allergen exposure in the lung, CD25-deficient CD4+ T cells have been shown to expand normally in the secondary lymphoid organs but migrate to the lung less effectively than WT CD4+ T cells, but the impact of CD25 deficiency on their Th2

fate decision per se remains unclear[33]. To definitively address the role of IL-2R signaling in Th2 cell differentiation in vivo, we first neutralized IL-2 by intraperitoneally (i.p.) injecting an anti-IL-2 mAb S4B6-1, which blocks the binding of IL-2 to CD25[34,35], in WT mice immunized with OVA plus papain as in Supplementary Fig. 3a. While the IL-2 blockade had no major impact on cell divisions and induced only a modest decrease in numbers of OT-II cells (Supplementary Fig. 3b, c), it inhibited the

**Fig. 1 | Direct antigen presentation by CD301b⁺ DCs is required for Th2 cell differentiation. a** Experimental design. Control (*Mgl2*⁺/*Cre*) and CD301b^ΔMHCII^ mice were adoptively transferred with OT-II cells and immunized one day later with OVA plus papain (**b–e**), OVA plus alum (**f–h**), or OVA plus CpG (**i–k**) in the footpad. The draining popliteal LNs were harvested 7 (**b–h**) or 5 (**i–k**) days after the immunization and restimulated ex vivo with PMA and ionomycin. **b–k** Frequencies of OT-II cells that have undergone indicated number of cell divisions (**b, f, i**), numbers of OT-II cells (**c, g, j**), flow cytometry plots of IL-4 and CFSE (**d**), and frequencies of IL-4⁺ and IFNγ⁺ cells among the donor OT-II cells (**e, h, k**) are shown. **l–n** Control (*Mgl2*⁺/*Cre*) and CD301b^ΔMHCII^ mice were subcutaneously infected with *Nb* in the ear.

Cervical (Cer) and mediastinal (Med) LNs were harvested 7 days after the infection and restimulated ex vivo with PMA and ionomycin. Numbers of CD44⁺ CD4⁺ T cells (**l**), flow cytometry plots of IL-4 and IFNγ (**m**), and frequencies of IL-4⁺ cells among the CD44⁺ CD4⁺ T cells (**n**) are shown. Data represent means ± SEM of 12 control and 10 CD301b^ΔMHCII^ mice for (**b–e**), 8 control and 6 CD301b^ΔMHCII^ mice for (**f–h**), 4 control and 4 CD301b^ΔMHCII^ mice for (**i–k**), 7 control and 6 CD301b^ΔMHCII^ mice for (**l–n**), or show representative flow cytometric plot of at least two independent experiments (**d**). Statistical analyses were performed using a two-tailed Student's *t*-test. Source data are provided as a Source Data file.

differentiation of IL-4-producing Th2 cells without affecting their ability to induce IFNγ-producing Th1 cells (Supplementary Fig. 3d), suggesting that IL-2 is specifically required for Th2 differentiation. Notably, the selective requirement of IL-2 for Th2 differentiation is not specific to the type 2 immune environment, since, when treated with the anti-IL-2 mAb, mice transferred with OT-II cells expressing the IL-4-GFP reporter (4get;OT-II) and immunized with OVA plus FCA, which predominantly induces non-type 2 inflammation, also exhibited significantly reduced IL-4-GFP expression in the donor 4get;OT-II cells without major alteration in their proliferation or IFNγ production (Supplementary Fig. 3e–g).

Since the anti-IL-2 mAb S4B6-1 can either neutralize or potentiate IL-2R signaling in a dose-dependent manner[36,37], we next examined if the reduced, but not completely abrogated, CD25 upregulation in CD4⁺ T cells was responsible for the impaired Th2 differentiation in CD301b⁺ DC-depleted mice by transferring *Il2ra* (CD25)⁺/⁺, *Il2ra*⁺/⁻, or *Il2ra*⁻/⁻ OT-II cells into WT mice immunized with OVA plus papain (Supplementary Fig. 3h). As expected, CD25 expression in the *Il2ra*⁺/⁺ OT-II cells was reduced by approximately 50% when measured 16 h after the dLN entry, whereas IL-2 production was increased inversely with the *Il2ra* gene dosage likely due to the lack of the CD25-dependent negative feedback[38] (Supplementary Fig. 3i, j). Nevertheless, STAT5 phosphorylation was reduced by roughly 50% and nearly absent in the *Il2ra*⁺/⁻ and *Il2ra*⁻/⁻ OT-II cells, respectively (Supplementary Fig. 3k), suggesting that the CD25 expression level, rather than the IL-2 production, from OT-II cells is the primarily determinant of STAT5 phosphorylation in this model. The reduction of pSTAT5 in the *Il2ra*⁺/⁻ OT-II cells were associated with lower levels of GATA-3 expression 24 h after the priming (Supplementary Fig. 3l, m), indicating that the maximal IL-2R signaling is necessary for the early commitment to the Th2 cell fate. Five days after immunization with OVA plus papain without CD62L blockade (Supplementary Fig. 3n), >80% of *Il2ra*⁺/⁻ and *Il2ra*⁻/⁻ OT-II cells underwent cell divisions, though there were mild defects in the number of division cycles as well as the total numbers of the donor OT-II cells (Supplementary Fig. 3o, p). However, the divided *Il2ra*⁺/⁻ and *Il2ra*⁻/⁻ OT-II cells failed to produce IL-4 without showing a defect in IFNγ production (Supplementary Fig. 3q). Immunization with OVA plus FCA also resulted in a loss of IL-4 production in the *Il2ra*⁺/⁻ OT-II cells, but in this case with a trend for reduction in IFNγ production without significant impact on proliferation (Supplementary Fig. 3r–t).

While the above data suggest Th cell-intrinsic requirement of full CD25 expression for Th2 cell differentiation, the potentially elevated IL-2 levels in the dLN due to the increased production from the *Il2ra*⁺/⁻ and *Il2ra*⁻/⁻ OT-II cells (Supplementary Fig. 3i, j) may have activated bystander T regulatory (Treg) cells and indirectly suppressed Th cell differentiation as previously shown for self-reactive Th cells[39,40]. To further elucidate the role of IL-2R signaling in Th2 fate decision in vivo, we next co-transferred equal numbers of *Il2ra*⁺/⁺ and *Il2ra*⁺/⁻ or *Il2ra*⁻/⁻ OT-II cells into mice immunized with OVA plus papain, FCA, or CpG, so that the IL-2 availability in the dLN is normalized (Fig. 3a, d). The *Il2ra* gene dosage-dependent decrease and increase in IL-2 production and STAT5 phosphorylation, respectively, were also observed in this setting (Fig. 3b, c), but, unlike the individual transfer approach (Supplementary Fig. 3o), the *Il2ra*⁺/⁻ OT-II cells showed comparable expansion

to the co-transferred *Il2ra*⁺/⁺ counterpart in papain- and FCA-immunized mice while they were outcompeted by the *Il2ra*⁺/⁺ OT-II cells in CpG-immunized mice, whereas the *Il2ra*⁻/⁻ OT-II cells expanded significantly less than the *Il2ra*⁺/⁺ OT-II cells regardless of the adjuvant used (Fig. 3e–g). The reduced expansion was not due to poor survival (Fig. 3h), suggesting that it was instead due to impaired survival and/or recruitment of naïve OT-II cells to the dLN. Importantly, the loss of even a single copy of *Il2ra* in the donor OT-II cells consistently resulted in a dramatic reduction in IL-4 production, whereas its impact on IFNγ production was variable and less significant (Fig. 3i). When normalized to the *Il2ra*⁺/⁺ counterpart in the same host, a similar reduction of Th2 cells was observed in the *Il2ra*⁺/⁻ OT-II cells in both papain- and FCA-immunized mice (Fig. 3j). Likewise, there was no significant difference in the reduction in Th1 cells in the *Il2ra*⁺/⁻ OT-II cells between papain-, FCA- and CpG-immunized mice when normalized to the *Il2ra*⁺/⁺ OT-II cells in the same host (Fig. 3j). Furthermore, the impact of the complete loss of CD25 on both Th1 and Th2 cell differentiation of the *Il2ra*⁻/⁻ OT-II cells was also largely similar between different models when normalized to the *Il2ra*⁺/⁺ counterpart (Fig. 3j). These data indicate that, despite the substantial difference in the magnitude of Th1 and Th2 cell differentiation induced by different adjuvants, the impact of the loss of CD25 on their differentiation was similar between models and does not correlate with the magnitude of differentiation. Most importantly, the loss of CD25 consistently had a greater impact on Th2 cell differentiation than on Th1 cell differentiation regardless of the model (Fig. 3j). Collectively, these data indicate that the Th2 fate decision by antigen-specific CD4⁺ T cells in vivo requires a potent and cell-intrinsic IL-2R signaling and is more sensitive to a partial loss of CD25 than the Th1 cell differentiation.

**CD301b⁺ DC-derived IL-2 is required for the full CD25 upregulation and Th2 differentiation in antigen-specific CD4⁺ T cells**

Following TCR ligation, CD25 is induced in antigen-specific CD4⁺ T cells through two consecutive steps: the initial upregulation induced by the TCR signaling and the subsequent TCR-independent amplification of the *Il2ra* transcription by the IL-2-driven, CD25-dependent positive feedback[41,42]. While the activated CD4⁺ T cells themselves are generally considered as the critical source of IL-2 in this feedback loop[42–44], DCs can also produce IL-2 upon activation in both mice and humans[45–49], though its role in T cell priming in vivo remains elusive. In mice immunized with OVA plus papain and transferred with OT-II cells (Fig. 4a), we found that CD301b⁺ DCs in the dLN expressed significantly higher levels of IL-2 than those in the non-draining lymph node (ndLN), while the increase was not observed in other DC subsets (Fig. 4b and Supplementary Fig. 4a, b). To examine if CD301b⁺ DC-derived IL-2 plays a critical role in CD25 upregulation and Th2 fate decision in antigen-specific CD4⁺ T cells, we generated mice lacking IL-2 specifically in CD301b⁺ DCs (*Mgl2*⁺/*cre*;*Il2*^fl/fl^, CD301b^ΔIL-2^ mice)[50] (Supplementary Fig. 4c–f) and monitored the activation and differentiation of the donor OT-II cells upon immunization with OVA plus papain (Fig. 4c–k). Notably, blocking IL-2 in WT recipients resulted in reduced CD25 expression in the donor OT-II cells. Likewise, the donor OT-II cells failed to fully upregulate CD25 in CD301b^ΔIL-2^ mice even though nearly all the OT-II cells expressed CD69 (Fig. 4d–f), indicating the specific

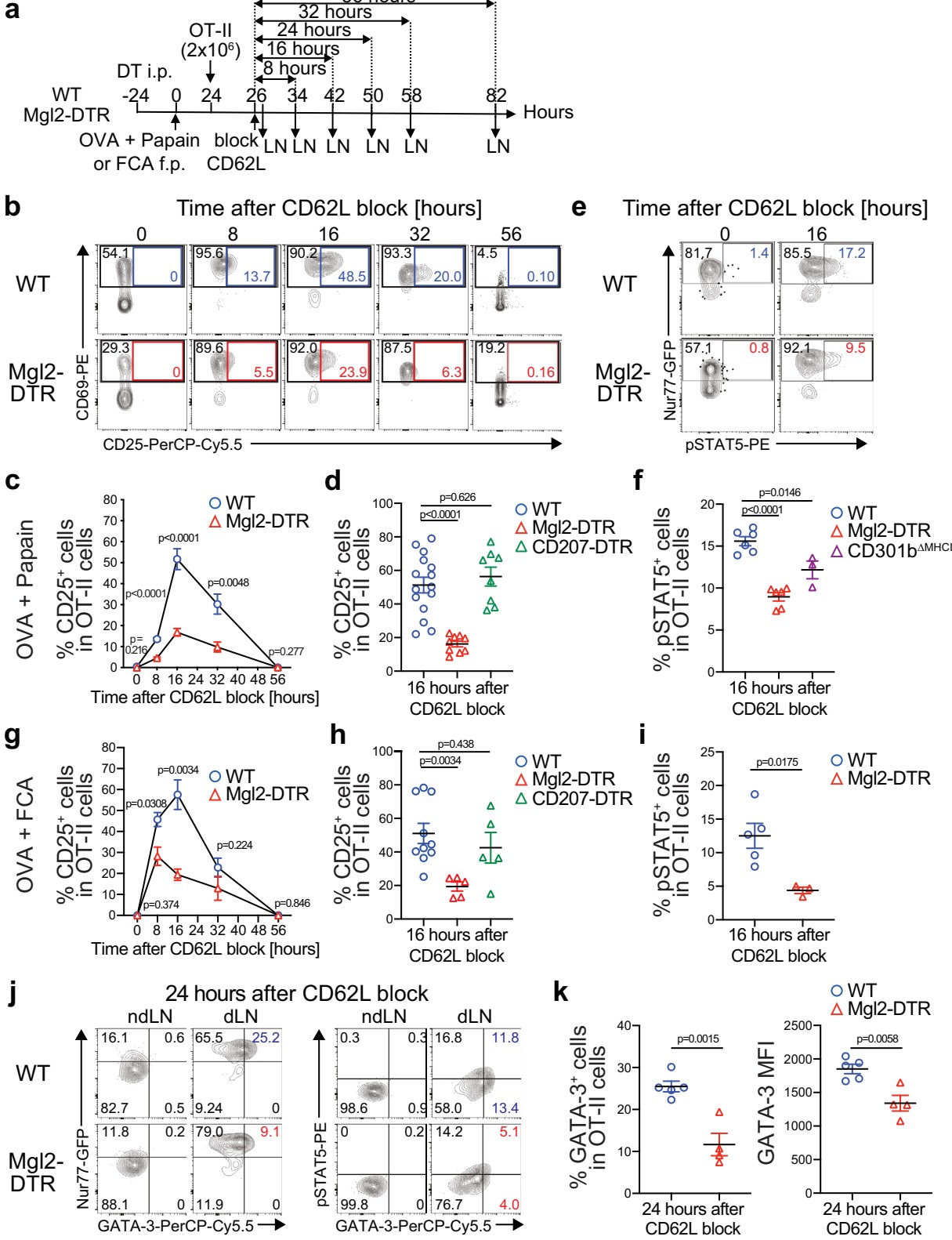

**a**

**b** Time after CD62L block [hours]

**c** OVA + Papain

**d**

**e** Time after CD62L block [hours]

**f**

**g** OVA + FCA

**h**

**i**

**j** 24 hours after CD62L block

**k**

requirement of CD301b[+] DC-derived IL-2 for fully upregulating CD25 in antigen-specific CD4[+] T cells. Furthermore, the differentiation of Th2 cells, monitored using the 4get;OT-II reporter cells, was abolished in the CD301b[ΔIL-2] recipients without affecting the cell cycle progression, expansion, or their differentiation into IFNγ-producing Th1 cells (Fig. 4g–k). In contrast, CD301b[+] DC-derived IL-2 was dispensable for the differentiation and homeostasis of DCs and Treg cells

(Supplementary Fig. 4f–h). Taken together, these results demonstrate that CD301b[+] DC-derived IL-2 is critical for inducing maximal CD25 upregulation and Th2 fate decision by antigen-specific CD4[+] T cells.

**CD301b[+] DC-derived CD25 is required for Th2 cell differentiation**
The above data suggest that CD301b[+] DCs provide IL-2 specifically to cognate CD4[+] T cells, but it remains unclear how the directed action of

**Fig. 2 | CD301b⁺ DCs are required for maximal CD25 expression and STAT5 activation in antigen-specific CD4⁺ T cells. a** Experimental design. OT-II cells were transferred into DT-treated Mgl2-DTR or WT mice 24 h after immunization with OVA plus papain (**b–f, j, k**) or OVA plus FCA (**g–i**) in the footpad. Two hours later, further homing of lymphocytes to LNs was blocked by injecting anti-CD62L mAb retro-orbitally. The dLNs were harvested at indicated time-points after the CD62L blockade. The donor OT-II cells were labeled by a congenic marker (CD45.1) or a cell tracer dye. OT-II cells expressing the Nur77-GFP reporter were used in (**e, f, i–k**). DT-treated CD207-DTR mice and CD301b^ΔMHCII mice (without DT) were used as additional groups of recipients where indicated. **b–k** Representative flow cytometric plots for indicated markers (**b, e, j**), frequencies of CD25⁺ cells (**c, d, g, h**), pSTAT5⁺ cells (**f, i**), and GATA-3⁺ cells and GATA-3 MFI (**k**) among the donor OT-II cells are shown. Data represent means ± SEM of 3–15 mice per group at each time point for (**c**) and (**g**), WT (*n* = 15), Mgl2-DTR (*n* = 9), and CD207-DTR (*n* = 8) for (**d**), WT (*n* = 6), Mgl2-DTR (*n* = 6), and CD301b^ΔMHCII (*n* = 3) for (**f**), WT (*n* = 10), Mgl2-DTR (*n* = 5), and CD207-DTR (*n* = 5) for (**h**), WT (*n* = 5) and Mgl2-DTR (*n* = 3) for (**i**), WT (n = 5) and Mgl2-DTR (*n* = 4) for (**k**), or show representative flow cytometry plots of at least two independent experiments (**b, e, j**). Statistical analyses were performed using a two-tailed Student's *t*-test. Source data are provided as a Source Data file.

IL-2 is protected from diffusion, which would lead to rapid consumption of IL-2 by bystander Treg cells rather than by cognate CD4⁺ T cells[39,40,43,44,51]. Previous studies have suggested that the diffusion of IL-2 is limited by the expression of CD25 on non-target cells[48,52–54]. We found that the expression of CD25 was higher in CD301b⁺ DCs than in CD301b⁻ DCs (Fig. 4l and Supplementary Fig. 4i). CD301b⁺ DCs also expressed higher levels of CD132 (IL-2Rγ) than CD301b⁻ DCs except for XCR1⁺ migDC1, but they express little, if any, CD122 (IL-2Rβ) (Supplementary Fig. 4j, k), suggesting that the IL-2R in CD301b⁺ DCs does not efficiently signal, as CD122 is required for the signaling[28]. Indeed, when the total LN cells from naïve WT mice were stimulated ex vivo with IL-2, pSTAT5 was detected in CD4⁺ T cells but not in DCs, while stimulation with GM-CSF, another potent STAT5 activator, induced pSTAT5 in DCs but not in CD4⁺ T cells (Supplementary Fig. 4l), further suggesting that CD25 in CD301b⁺ DCs does not promote IL-2 signaling *in cis*. Importantly, in mice lacking CD25 specifically in CD301b⁺ DCs (*Mgl2^+/Cre;Il2ra^fl/fl*, CD301b^ΔCD25 mice)[55,56] (Supplementary Fig. 4m, n), differentiation of the donor OT-II cells into Th2 cells was significantly impaired without affecting their expansion or Th1 cell differentiation or the number of each DC subset (Fig. 4m–q, Supplementary Fig. 4o). The lack of CD25 in CD301b⁺ DCs resulted in a reduction of pSTAT5 specifically in the donor OT-II cells but not in the endogenous Treg cells (Fig. 4r–t), suggesting that CD301b⁺ DC-intrinsic CD25 expression facilitates the directed action of IL-2 toward cognate CD4⁺ T cells. Collectively, these results indicate that CD301b⁺ DCs promote Th2 cell differentiation by reinforcing the availability of IL-2 for cognate CD4⁺ T cells by the DC-intrinsic expression of CD25.

## CD40 stimulation induces IL-2 production specifically by CD301b⁺ DCs

To further explore the mechanism for the Th2 cell fate instruction by CD301b⁺ DCs, we performed CITEseq, a single-cell RNA sequencing (scRNAseq) technique with DNA-barcoded mAbs for simultaneously analyzing the expression of mRNA and cell surface markers[57], on MHCII^hi migratory DCs isolated form skin-dLNs of naïve and papain-immunized mice one day after immunization. Unsupervised clustering analysis of the cell surface markers identified 7 distinct clusters composed of 3 major DC subsets, including LC (cluster 4: CD172a^lo CD326^hi XCR1⁻), cDC1s (cluster 5: CD172a⁻ CD326^int XCR1⁺), and cDC2s (clusters 1, 2, 3, 6 and 7: CD172a^hi CD326⁻ XCR1⁻) (Fig. 5a). The cDC2s were divided into three subpopulations, CD301b^lo, CD301b^int, or CD301b^hi DCs, based on CD301b expression levels (Fig. 5b). Alternatively, the cDC2s were further divided into five subclusters based on their mRNA expression profile, with the subclusters 2 and 5 mainly composed of naïve cDC2s, while subclusters 1, 3, and 4 predominantly derived from cDC2s in immunized mice (Fig. 5a, c). While the CD301b expression alone did not identify a single cDC2 subcluster in either naïve or immunized mice, cells in the cDC2 subcluster 1, 2 and 3 were more enriched in CD301b^int and CD301b^hi DCs, whereas those in the subcluster 4 and 5 were more abundant in CD301b^lo DCs (Fig. 5b, c). Consistent with the intracellular staining data (Supplementary Fig. 4b), *Il2* mRNA was detected in all DC subsets including CD301b^hi DCs (Fig. 5d). The Ingenuity Pathway Analysis[58] of the genes differentially expressed in CD301b^hi DCs compared with all other DC subsets in papain-immunized mice identified several pathways including the CD40 pathway, which was enriched in CD301b^hi DCs in both naïve and papain-immunized mice (Fig. 5e, f). Indeed, CD301b⁺ DCs expressed higher levels of CD40 compared to CD301b⁻ DCs in both dLN and ndLN of papain-immunized mice (Fig. 5g, h). Importantly, stimulation of CD40 by i.p. administration of an agonistic anti-CD40 mAb FGK4.5[59–61] induced IL-2 production in CD301b⁺ DCs along with CD80, CD86, and CD25 (Fig. 5i, j). Notably, the CD40-induced IL-2 production was specific to CD301b⁺ DCs among DC subsets (Fig. 5i). Thus, CD40 ligation stimulates IL-2 production and upregulation of costimulatory molecules in CD301b⁺ DCs.

## CD301b⁺ DC-intrinsic CD40 is required for Th2 cell differentiation

Previous studies have shown the requirement of CD40 in Th2 cell differentiation[62,63], but the underlying mechanism is unknown. Indeed, the 4get;OT-II cells primed with OVA plus papain in CD40-deficient mice expressed significantly lower levels of GFP (reporting *Il4*)[64] compared to those in WT mice, whereas their expansion, cell division, and differentiation into Th1 cells remained comparable (Supplementary Fig. 5a–c). Moreover, a similar reduction was observed for Th2 cells but not for Th1 cells when the OT-II cells were primed in mixed bone marrow (BM)-chimeric (BMC) mice reconstituted with a 1:1 mixture of MHCII-deficient and CD40-deficient BM cells, in which cells expressing MHCII do not express CD40, while cells expressing CD40 cannot present antigens to CD4⁺ T cells (Fig. 6a–c). Intriguingly, unlike OT-II cells primed in CD40-deficient mice, the cell division and expansion of OT-II cells were also impaired in MHCII⁻/⁻:CD40⁻/⁻ BMC mice (Supplementary Fig. 5d, e), suggesting that the partial loss of MHCII-dependent cognate TCR stimulation in the BMC mice had an additional negative impact on priming. These data indicate the requirement of the concurrent CD40 signaling and antigen presentation in the same cells specifically for Th2 fate instruction.

We next reconstituted lethally irradiated WT mice with a 1:1 mixture of CD40⁻/⁻ and Mgl2-DTR BM cells (Fig. 6d). In the resulting mice (CD301b^ΔCD40 BMC), only CD40⁻/⁻ cells remained intact in the CD301b⁺ compartment upon DT injection, while all CD301b⁻ compartments maintained the 1:1 chimerism (Fig. 6e). Although the MHCII expression in CD40⁻/⁻ CD301b⁺ DCs was comparable to that in WT CD301b⁺ DCs (Fig. 6f), the OT-II cells primed with OVA plus papain in the CD301b^ΔCD40 BMC mice had significantly reduced Th2 cells but not Th1 cells (Fig. 6g). Similarly to the MHCII⁻/⁻:CD40⁻/⁻ BMC mice (Supplementary Fig. 5d, e), there was a slight but significant reduction in the cell division and expansion of OT-II cells in the CD301b^ΔCD40 BMC mice (Supplementary Fig. 5f, g), suggesting that the loss of 50% of CD301b⁺ DCs in these mice partially attenuated the priming itself. Collectively, these data indicate that CD301b⁺ DC-intrinsic CD40 is required for Th2 cell fate instruction.

## CD301b⁺ DCs are a sufficient DC subset for Th2 cell fate instruction

The data thus far indicate that direct antigen presentation, CD40 expression, and IL-2 production by CD301b⁺ DCs are all *required* for Th2 cell fate instruction. We previously showed that adoptively

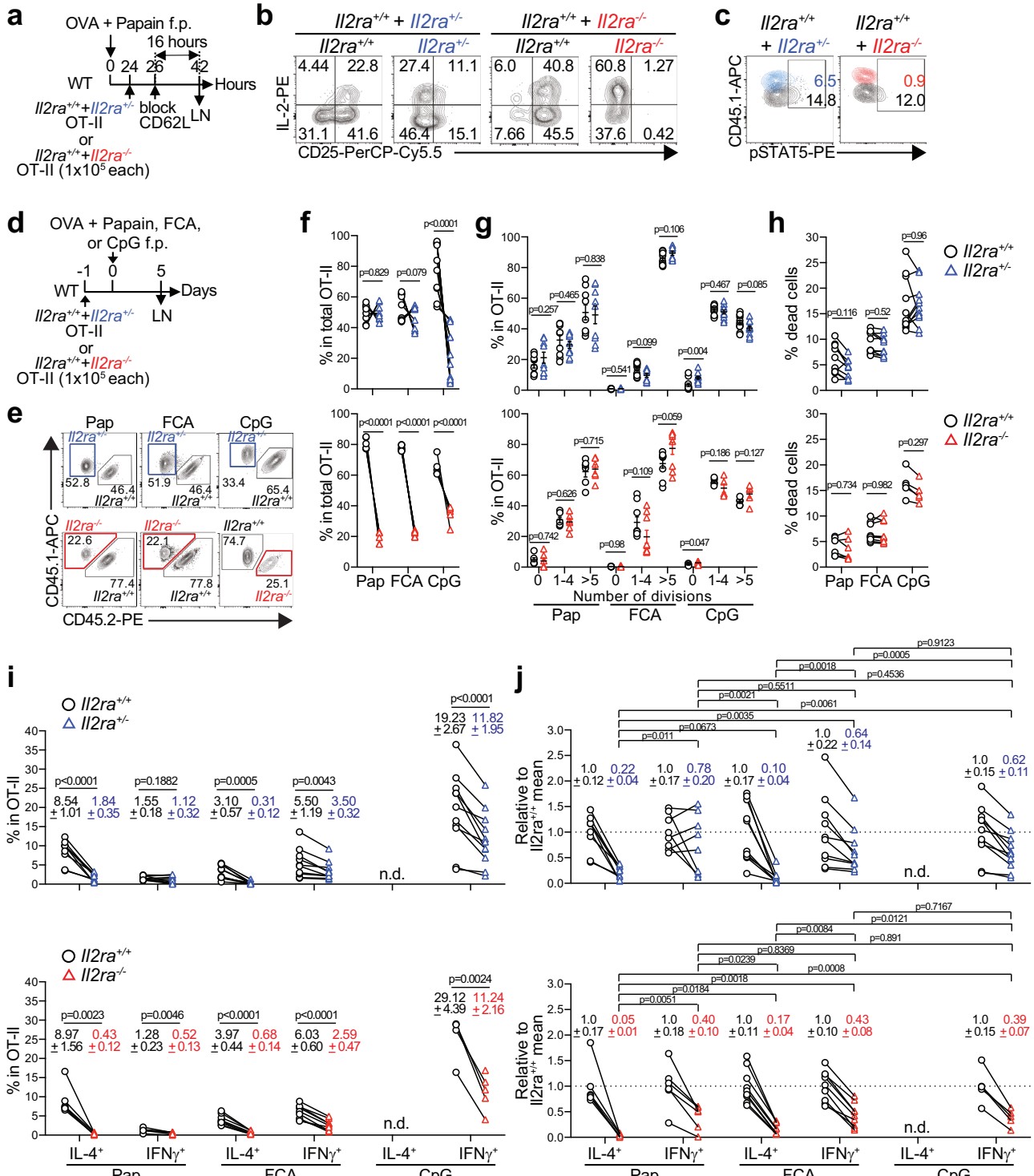

**Fig. 3 | Th2 cell differentiation relies more stringently on IL-2R signaling than Th1 cells. a–c** A 1:1 mixture of CFSE-labeled and congenically marked *Il2ra+/+* and *Il2ra+/–* OT-II cells, or *Il2ra+/+* and *Il2ra–/–* OT-II cells were co-transferred into WT mice that were immunized and treated as in (**a**). Representative flow cytometry plots of CD25 and IL-2 (**b**) and pSTAT5 (**c**) among the donor OT-II cells are shown. **d–j** WT mice were co-transferred with a 1:1 mixture (1 × 10⁵ cells each) of congenically distinct *Il2ra+/+* and *Il2ra+/–*, or *Il2ra+/+* and *Il2ra–/–* OT-II cells, and treated and immunized with OVA mixed with papain (*n* = 9 for *Il2ra+/+* and *Il2ra+/–* OT-II, *n* = 6 for *Il2ra+/+* and *Il2ra–/–* OT-II), FCA (*n* = 10 for *Il2ra+/+* and *Il2ra+/–* OT-II, *n* = 9 for *Il2ra+/+* and *Il2ra–/–* OT-II), or CpG (*n* = 11 for *Il2ra+/+* and *Il2ra+/–* OT-II, *n* = 5 for *Il2ra+/+* and *Il2ra–/–* OT-II) as indicated (**d**). Flow cytometry plots of CD45.1 and CD45.2 expression by gated CD45.1⁺ OT-II cells (**e**), paired frequencies of indicated OT-II

cells among the total OT-II cells within individual host dLNs (**f**), frequencies of indicated OT-II cells that have undergone indicated number of cell divisions (**g**), frequencies of Zombie-Aqua⁺ dead cells among indicated OT-II cells (**h**), paired relative frequencies of IL-4⁺ and IFNγ⁺ cells among the CD44⁺ CFSE^lo OT-II cells within individual host dLNs (**i**). In (**j**), frequencies of IL-4⁺ and IFNγ⁺ cells of *Il2ra+/–* or *Il2ra–/–* OT-II cells in (**i**) were normalized to the mean frequency of the *Il2ra+/+* OT-II counterpart. In (**h–j**), each connecting line indicates OT-II cells of two different genotypes isolated from the same host. Data represent means ± SEM (**f–j**) or show representative flow cytometry plots of at least two independent experiments (**b, c, e**). Statistical analyses were performed using a two-tailed Student's *t*-test (**g, j**) or two-tailed paired *t*-test (**f, h, i**). n.d. not detectable. Source data are provided as a Source Data file.

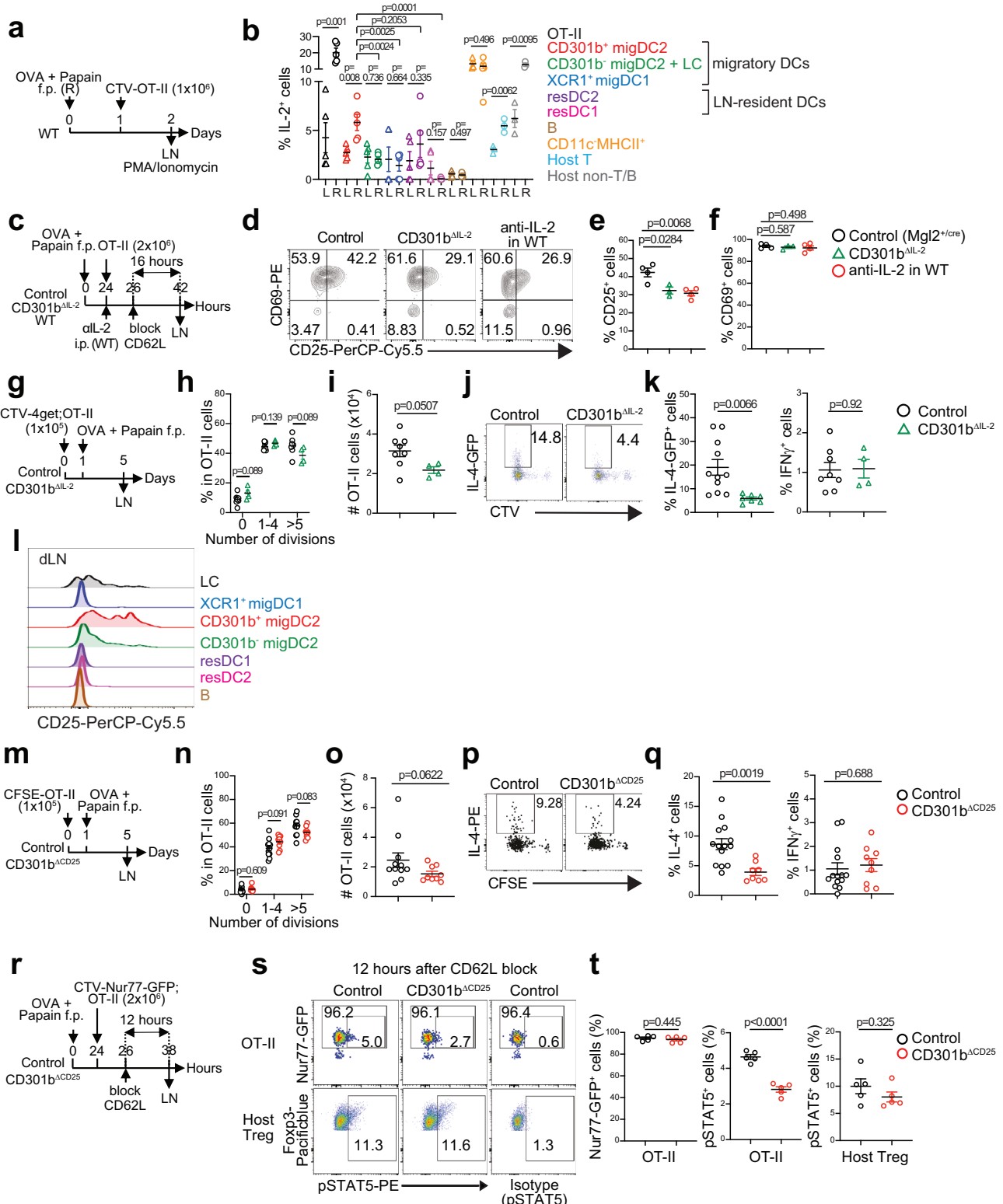

transferring CD301b⁺ DCs sorted from papain-immunized mice and pulsed ex vivo with an antigen is not sufficient to induce Th2 cell differentiation in naïve recipients[13]. This may reflect the involvement of multiple DC subsets for Th cell differentiation as suggested by others[65,66] or may be due to the requirement of cells outside of the DC compartment that were not fully recapitulated in the recipients as the recipients had never been directly exposed to the adjuvant in such experiments. To address if CD301b⁺ DCs are *sufficient* and no other DC subsets are required for Th2 cell fate instruction, we took advantage of

mice expressing a Cre-removable DTR cassette in CD11c⁺ cells (CD11c-dlDTR) crossed with the Mgl2-Cre strain (Mgl2-Cre;CD11c-dlDTR mice), in which CD301b⁺ DCs are protected from DT-induced depletion while CD301b⁻ DCs are susceptible to DT[20]. Although the number of CD301b⁺ cells in these mice is underestimated due to the loss of CD301b protein expression from the *Mgl2^cre* allele[20], the use of the inducible tdTomato reporter concurrently with these mice (Mgl2-Cre;CD11c-dlDTR;R26^LSL-iTom) confirmed subset-specific protection of Cre-expressing *Mgl2* (CD301b)⁺ DCs from DT-induced depletion in the skin-dLNs

**Fig. 4 | CD301b$^+$ DC-derived IL-2 and CD25 are required for the maximal IL-2R signaling and Th2 cell differentiation of antigen-specific CD4$^+$ T cells. a, b** WT mice were immunized with OVA plus papain in the right (R) footpad and transferred with Cell Tracer Violet (CTV)-labeled OT-II cells. The dLNs (R) and left (L) ndLNs were harvested and restimulated with PMA and ionomycin (**a**). Frequencies of IL-2-producing cells in each cell type were defined as in Supplementary Fig. 4a and enumerated (**b**). **c–f** Control (*Mgl2$^{+/Cre}$*), CD301b$^{\Delta IL-2}$, and WT mice were immunized and treated as indicated (**c**). The WT recipients were also treated with anti-IL-2 neutralizing mAb. Representative flow cytometry plots of CD69 and CD25 (**d**), frequencies of CD25$^+$ (**e**) and CD69$^+$ (**f**) cells among the donor OT-II cells in the dLN are shown. **g–k** Control and CD301b$^{\Delta IL-2}$ mice were immunized and treated as in (**g**). The dLNs cells were restimulated with PMA and ionomycin. Frequencies of OT-II cells with the indicated number of cell divisions (**h**), numbers of OT-II cells (**i**), representative flow cytometry plots of IL-4-GFP reporter and CTV dilution (**j**), and frequencies of IL-4-GFP$^+$ Th2 and IFNγ$^+$ Th1 cells among the donor OT-II cells (**k**) are shown. **l** Representative flow cytometry histograms showing CD25 (IL-2Ra) expression in indicated DC subsets. WT mice were immunized with OVA plus papain in the footpad and the draining popliteal LNs were harvested 24 h after the immunization. **m–q** As in (**g**), but CD301b$^{\Delta CD25}$ mice were used as recipients (**m**). Frequencies of OT-II cells with the indicated number of cell divisions (**n**), numbers of OT-II cells (**o**), representative flow cytometry plots of IL-4 and CFSE dilution (**p**), and frequencies of IL-4$^+$ Th2 and IFNγ$^+$ Th1 cells among the donor OT-II cells (**q**) are shown. **r–t** Control and CD301b$^{\Delta CD25}$ mice were immunized and treated as in (**r**). Representative flow cytometry plots of pSTAT5 and Nur77-GFP in OT-II cells or pSTAT5 and Foxp3 among CD25$^+$ Foxp3$^+$ CD4$^+$ Treg cells (**s**), frequencies of Nur77-GFP$^+$ or pSTAT5$^+$ among OT-II cells, and pSTAT5$^+$ among Treg cells (**t**) in the dLN are shown. Data represent means ± SEM of WT (n = 5) for (**b**), control (n = 4), CD301b$^{\Delta IL-2}$ (n = 3), and WT treated with anti-IL-2 mAb (n = 4) for (**e, f**), control (n = 8) and CD301b$^{\Delta IL-2}$ (n = 4) for (**h, i, k**), control (n = 11) and CD301b$^{\Delta CD25}$ (n = 9) for (**n, o, q**), control (n = 5) and CD301b$^{\Delta CD25}$ (n = 5) for (**t**) or show representative flow cytometry plots of at least two independent experiments (**d, j, p, s**). Statistical analyses were performed using a two-tailed Student's *t*-test (**b, e, f, h, I, k, n–q, t**). Source data are provided as a Source Data file.

---

(Supplementary Fig. 5h−m). In line with the tdTomato expression pattern in the *Mgl2$^{+/Cre}$;R26$^{LSL-iTom}$* mice (Supplementary Fig 1a, b), the majority of the protected DCs expressed PD-L2, a marker known to be correlated with CD301b expression in CD172a$^+$ cDC2s[12,19], whereas all other DCs, including PD-L2$^-$ migDC2s, LCs, XCR1$^+$ migDC1s, and LN-resident DCs were effectively depleted (Supplementary Fig. 5i−m). Importantly, OT-II cells primed in the Mgl2-Cre;CD11c-dlDTR mice differentiated normally into Th1 and Th2 cells under both Th2 (papain) and non-Th2 (FCA) immunization conditions (Fig. 6h–j), though modest reduction in their expansion was observed in both conditions (Supplementary Fig. 5n−q). These results indicate that, while DCs other than CD301b$^+$ DC subset may be required for the maximal OT-II cell expansion, CD301b$^+$ DCs are both required and sufficient for Th2 cell fate instruction in these immunization conditions.

## CD301b$^+$ DCs instruct Th2 cell fate on CD4$^+$ T cells through the DC-intrinsic CD40−IL-2 axis

To further elucidate if the sufficiency of CD301b$^+$ DCs for Th2 cell fate instruction is based on their cell-intrinsic CD40−IL-2 axis, we next utilized anti-CD154 (CD40L) blocking mAb (MR-1) or anti-CD40 (FGK4.5) agonistic mAb, both of which serve to inhibit physical interaction between CD40 and CD154[60,61,67,68]. Blockade of CD154 in WT mice or genetic deficiency of CD40, but not stimulation of CD40, resulted in impaired upregulation of CD25 without affecting the CD69 expression in OT-II cells primed with OVA plus papain (Fig. 6k–m). Accordingly, the blockade of CD154 blunted Th2 cell differentiation of the donor OT-II cells in WT mice, whereas stimulation of CD40 restored Th2 cell differentiation even when CD154 was blocked, indicating that CD40 stimulation is sufficient to drive Th2 cell differentiation even in the absence of CD154 signaling in CD4$^+$ T cells (Fig. 6n, o). However, the CD40 stimulation failed to restore Th2 cell differentiation when IL-2 was additionally neutralized, suggesting that IL-2 is responsible for the Th2 fate instruction downstream of CD40 stimulation (Fig. 6o). Furthermore, the CD40 stimulation did not restore Th2 cell differentiation of the donor OT-II cells in CD301b$^+$ DC-depleted mice, CD301b$^{\Delta IL-2}$ mice, or in CD301b$^{\Delta MHCII}$ mice (Fig. 6o), indicating that the CD40 stimulation induces Th2 cell differentiation by inducing IL-2 production from the CD301b$^+$ DCs that are in cognate interaction with CD4$^+$ T cells. We observed no major impact on proliferation of the donor OT-II cells in any of the conditions (Fig. 6p). Notably, the CD40 stimulation significantly increased the frequencies of IFNγ$^+$ Th1 OT-II cells in WT mice (Supplementary Fig. 5r). Unlike CD40-induced Th2 cell differentiation, a trend for an increase in Th1 differentiation, though statistically insignificant, was also observed in CD301b$^+$ DC-depleted, CD301b$^{\Delta IL-2}$, or in CD301b$^{\Delta MHCII}$ mice upon stimulation of CD40, suggesting that the stimulation of CD40 in CD301b$^-$ antigen-presenting cells can promote Th1 cell differentiation.

Collectively, these data indicate that MHCII- and CD40-dependent interaction between CD301b$^+$ DCs and antigen-specific CD4$^+$ T cells and the resultant production of IL-2 from CD301b$^+$ DCs play a decisive role in Th2 cell fate instruction.

## CD301b$^+$ DC-intrinsic CD40−IL-2 axis skews CD4$^+$ T cells toward non-Tfh effector fate

We have shown previously that the depletion of CD301b$^+$ DCs in the Mgl2-DTR mice results in the expansion of Tfh cells in addition to impaired Th2 cell differentiation[19]. This Tfh expansion is due at least partially to the loss of PD-L1-dependent suppression of Tfh cells by CD301b$^+$ DCs[19], but it remains unclear if CD301b$^+$ DCs suppress the initial fate decision of antigen-specific CD4$^+$ T cells into Tfh cells. During CD4$^+$ T cell priming, Tfh cells downregulate PSGL1 prior to upregulating CXCR5[69]. Unlike CD301b$^+$ DC-depleted Mgl2-DTR mice[19], there was no significant increase in mature CXCR5$^+$ PD-1$^+$ Tfh cells within both OT-II and endogenous CD4$^+$ T cell compartments in CD301b$^{\Delta MHCII}$ mice immunized with OVA plus papain (Fig. 7a, b), suggesting that the expansion of mature CXCR5$^+$ PD-1$^+$ Tfh cells in CD301b$^+$ DC-depleted mice is mainly driven by the loss of antigen-independent suppression mechanism such as PD-L1-mediated suppression of Tfh cells[19]. However, the PSGL1$^{lo}$ PD-1$^+$ 'immature' Tfh fraction was significantly increased in both OT-II and endogenous CD4$^+$ T cells in CD301b$^{\Delta MHCII}$ mice, suggesting that more CD4$^+$ T cells are poised to the Tfh cell fate in those mice (Fig. 7c, d).

CD4$^+$ T cell-intrinsic IL-2R signaling suppresses Tfh cell differentiation while favoring the differentiation of non-Tfh effector Th cells[70–73], but the role of T cell-extrinsic source of IL-2 in this bifurcation is unknown. In WT mice transferred with OT-II cells, the blockade of IL-2 by anti-IL-2 mAb resulted in a significant increase in the differentiation of PSGL1$^{lo}$ PD-1$^+$ CD4$^+$ T cells upon immunization with OVA plus papain (Fig. 7e, f). Notably, the differentiation of the PSGL1$^{lo}$ PD-1$^+$ CD4$^+$ T cells was also increased in OT-II cells primed in CD301b$^{\Delta CD40}$ BMC mice and CD301b$^{\Delta IL-2}$ mice compared with those primed in respective control mice, which was also observed in their endogenous CD4$^+$ T cells (Fig. 7g–l). Interestingly, unlike in CD301b$^{\Delta MHCII}$ mice (Fig. 7a, b), the endogenous, but not OT-II, CXCR5$^+$ PD-1$^+$ mature Tfh cells were also increased in the CD301b$^{\Delta CD40}$ BMC mice compared with the WT:Mgl2-DTR mixed BMC control mice (Fig. 7i, j), suggesting that the depletion of 50% of CD301b$^+$ DCs in the CD301b$^{\Delta CD40}$ BMC mice, in combination with the lack of CD40 in the remaining CD301b$^+$ DCs, had more pronounced impact on Tfh cell suppression than the CD301b$^+$ DC-specific MHCII deficiency alone. Lastly, along with the downregulation of PSGL1, the expression of Bcl-6, the master regulator of Tfh cell differentiation[74], was also increased in the PD-1$^+$ fraction in OT-II cells and showed a trend for an increase in endogenous CD4$^+$ T cells in CD301b$^{\Delta IL-2}$ mice (Fig. 7k–n), further suggesting that more CD4$^+$ T cells are poised for Tfh

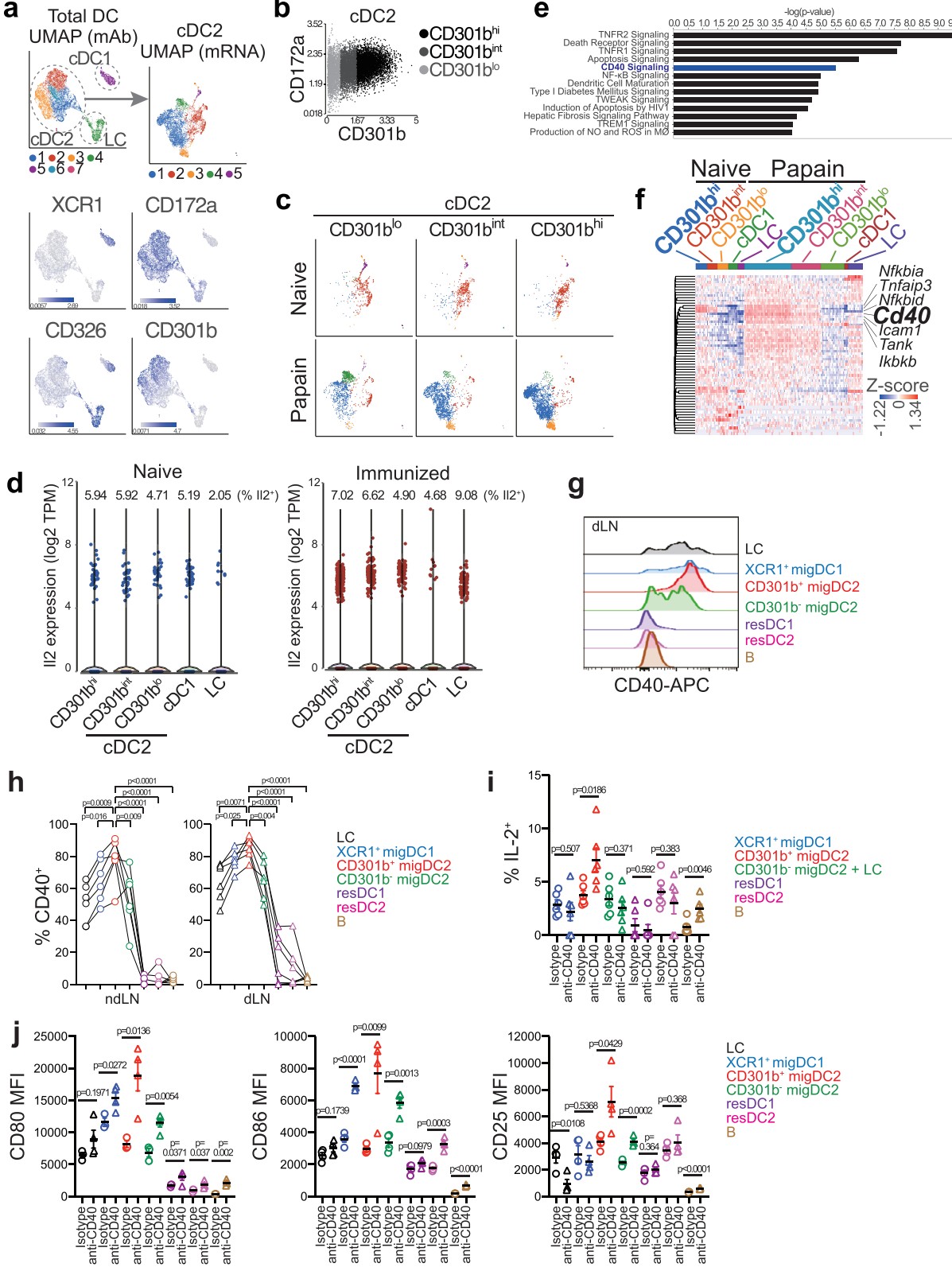

differentiation in these mice. Thus, in addition to driving the differentiation toward Th2 cell fate, these data indicate that the CD301b+ DC-intrinsic CD40–IL-2 axis also limits the commitment to Tfh cell fate and promotes non-Tfh effector Th cell fate. Taken together, these results demonstrate the role of CD301b+ DC-derived IL-2 in priming and effector fate bifurcation of CD4+ T cells (Fig. 8).

## Discussion

Th2 cells play a fundamental role in type 2 adaptive immunity such as host protection against helminth parasites and pathogenesis of allergic diseases. The transcription factor GATA-3 has long been known as the cell-intrinsic master regulator for the Th2 cell identity, but the mechanism of how the Th2 cell fate is instructed by exogenous cues in

**Fig. 5 | CD40 stimulation induces IL-2 production by CD301b⁺ DCs. a–f** MHCII^hi migratory DCs (including LCs) were sorted from the skin-dLNs of naïve mice and mice immunized with OVA plus papain for one day and labeled with DNA-barcoded mAbs for CD8α, CD11b, CD11c, CD24, CD64, CD103, CD172a, CD301b, CD326, Ly6C, MHCII and XCR1 for CITEseq analysis. Uniform Manifold Approximation and Projection (UMAP) plots for the seven distinct DC clusters separated by the cell surface marker expression (left) and the five cDC2 subclusters identified based on the mRNA expression (right) are shown (**a**). Clusters of cDC1, cDC2, and LCs were identified by the expression of XCR1, CD172a, and CD326, respectively, as indicated. cDC2 cells were divided into three subpopulations based on the cell surface CD301b expression as in (**b**) and projected onto the UMAP plots (**c**). *Il2* mRNA expression in indicated DC subsets in skin-dLNs of naïve and immunized mice is shown in (**d**). The *y*-axis indicates the log2 transcripts per million (TPM) and each dot represents a single cell. Ingenuity Pathway Analysis identified the CD40 pathway as one of the differentially expressed pathways in CD301b^hi cDC2s compared with all other DC populations in papain-immunized dLNs (**e**). The heatmap in (**f**) shows the expression of each gene in the CD40 pathway in each sequenced cell.

**g, h** Expression of CD40 in each DC subset in the dLN and ndLN of WT mice immunized with papain in the footpad for one day. Flow cytometry histograms in the dLN (**g**) and frequencies of CD40⁺ cells among each DC subset (*n* = 6) (**h**) are shown. In (**h**), lines connect values for the same mouse. **i, j** WT mice were i.p. injected with anti-CD40 agonistic or isotype control mAbs. Popliteal LNs were harvested 24 h after the injection and stimulated ex vivo with PMA and ionomycin for intracellularly staining IL-2. Alternatively, the cell surface expression of CD80, CD86, and CD25 was examined by flow cytometry. Frequencies of IL-2⁺ cells among indicated subsets (*n* = 6 for isotype and *n* = 6 for anti-CD40 mAb) (**i**) and MFI of CD80 (*n* = 3 for isotype and *n* = 6 for anti-CD40 mAb), CD86 (*n* = 4 for isotype and *n* = 4 for anti-CD40 mAb), and CD25 (*n* = 4 for isotype and *n* = 4 for anti-CD40 mAb) in indicated populations (**j**) are shown. Data represent means ± SEM (**h**–**j**), or show representative flow cytometry plots of at least two independent experiments (**g**). Statistical analyses were performed using the right-tailed Fisher's Exact Test with Benjamini-Hochberg multiple-testing correction (**e**), two-tailed paired *t*-test (**h**) or two-tailed Student's *t*-test (**i**–**j**). Source data are provided as a Source Data file.

vivo remains unclear. Although IL-4 is commonly used to induce Th2 cell differentiation in cultured CD4⁺ T cells, there is no clear evidence that DCs produce IL-4 to instruct the Th2 cell fate[75], and the expression of IL-4 receptor in CD4⁺ T cells is not strictly required for their Th2 fate decision in vivo[76]. The present study reveals that the DC-intrinsic CD40–IL-2 axis in CD301b⁺ DCs plays a crucial role in Th2 cell fate instruction by quantitatively regulating the IL-2R signaling in CD4⁺ T cells. Upon encountering antigen-specific naive CD4⁺ T cells, CD301b⁺ DCs not only present the antigen to stimulate the TCR for timely activation but also provide IL-2 to T cells upon CD40 ligation to induce full upregulation of CD25, which is required specifically for Th2 cell differentiation, as Th2 cells rely on the IL-2R signaling for their differentiation more stringently than Th1 cells. Furthermore, the CD301b⁺ DC-derived IL-2 also limits the differentiation into Tfh cells, favoring the commitment to non-Tfh effector Th cells altogether. These data demonstrate that CD301b⁺ DC-derived IL-2 plays a decisive role in Th cell fate instruction and highlight the importance of DC-dependent quantitative control of IL-2R signaling in CD4⁺ T cells for the commitment to Th2 cell fate.

Originally identified as the T cell growth factor secreted by activated T cells themselves[77,78], IL-2 plays a multifaceted role in the proliferation, differentiation, and survival of CD4⁺ T cells. Although recombinant IL-2 is often added as a cell culture supplement to activated CD4⁺ T cells to maintain the survival of 'Th0' cells without skewing them toward the Th1 or Th2 status, previous studies have shown that CD4⁺ T cell-intrinsic production and sensing of IL-2 inhibits the differentiation of Th17 and Tfh cells[70–72,79,80], while it is required for the differentiation of Th1 and Th2 cells[30,31,72,81–84]. In Th2 cell differentiation, STAT5, the main signaling component of the IL-2R, has been shown to directly bind to the *Il4*, *Il4ra*, and *Gata3* loci and promote their expression[30,83,84], leading to the activation of the IL-4–IL-4Rα–STAT6 positive feedback loop to accelerate the differentiation of Th2 cells and dictate their identity[85,86]. Accordingly, we have shown previously that CD301b⁺ DCs are required for the upregulation of IL-4Rα in antigen-specific CD4⁺ T cells during the priming[20]. However, STAT5 also promotes the expression of Th1-associated genes such as *Il12rb*, *Tbx21*, and *Ifng* in the context of Th1 cell differentiation[31,87]. Our data indicate that, while both Th1 and Th2 cell differentiation ultimately requires CD25 expression as shown in previous studies[29,72,88,89], Th2 cell differentiation is more sensitive to a partial loss of CD25. These observations suggest that quantitative differences in the IL-2R signaling during CD4⁺ T cell priming lead to qualitative differences in Th cell fate decision.

In line with the Th cell fate regulation by quantitative control of IL-2R signaling, the availability of IL-2 is tightly regulated in the LN microenvironment. Studies have shown that IL-2 produced by antigen-specific CD4⁺ T cells is rapidly quenched by Treg cells and/or DCs to prevent aberrant activation of self-reactive clones and to promote Tfh

cell differentiation[39,40,43,44,51,53]. However, since the IL-2 in CD4⁺ T cells functions predominantly in a paracrine manner[73], it is unclear how the IL-2R signaling overshoots the threshold in rare antigen-specific CD4⁺ T cells immediately after priming. Our data indicate that CD301b⁺ DCs are a critical source of IL-2 early after CD4⁺ T cell priming, especially when CD4⁺ T cells require the maximal IL-2R signaling for Th2 cell differentiation. This directed action of IL-2 is facilitated by the expression of CD25 by CD301b⁺ DCs. These data suggest that IL-2 may be directly trans-presented to cognate CD4⁺ T cells through the membrane-bound CD25 as previously suggested in other contexts[48,54], though the precise mechanism needs further investigation. Importantly, IL-2 production in CD301b⁺ DCs is induced by CD40, which needs to have coincided with MHCII-dependent cognate interaction for the Th2 cell fate instruction. Thus, CD301b⁺ DCs appear to 'kick-start' the IL-2R signaling in CD4⁺ T cells by directly handing IL-2 over to the cognate clones, thereby minimizing the IL-2 quenching effect by surrounding Treg cells. Interestingly, previous studies have shown the production of IL-2 by DCs at the interface between DCs and CD4⁺ T cells in vitro[46,48], suggesting that direct IL-2 delivery by CD301b⁺ DCs to CD4⁺ T cells through the immunological synapse ensures timely and confined consumption of IL-2 by the cognate clones.

While our data in CD301b⁺ DC-enriched Mgl2-cre;CD11c-dlDTR mice suggest that CD301b⁺ DCs are sufficient for inducing Th2 cell differentiation, the relatively normal Th1 cell differentiation in those mice, despite modestly impaired expansion, also indicates that the cognate interaction with CD301b⁺ DCs alone does not constrain the fate of CD4⁺ T cells to the Th2 status. Further, CD301b⁺ DCs are required for the maximal IL-2R signaling even under non-Th2 immunization conditions with FCA or CpG, but, unlike Th2 cell differentiation, it seems to be dispensable for the commitment to Th1 cell fate. Notably, in CD4⁺ T cells stimulated with cognate peptides in vitro, the dose of the peptide has been shown to correlate with the Th1-to-Th2 ratio while inversely correlating with STAT5 activation due to inhibition of STAT5 by ERK, a signal downstream of the TCR, implicating that Th1 cells only require a lower amount of activated STAT5 for their differentiation while Th2 cell differentiation requires more potent STAT5 activation[30]. Moreover, a recent study found that IL-6 induced by antigen-independent inflammation dampens Th2 and promotes Th17 cell differentiation by prematurely down-regulating CD25 expression in activated CD4⁺ T cells[90]. Thus, the non-Th2 fate appears to be associated with excess TCR signaling and/or inflammation that results in reduced IL-2R signaling. We reported recently that CD301b⁺ DCs in the LNs are located in closer proximity to high endothelial venules and have earlier access to incoming naive CD4⁺ T cells than LCs and cDC1s, which makes them critical for timely priming of antigen-specific CD4⁺ T cells[20]. Given that LCs and cDC1s encounter CD4⁺ T cells in a delayed timing from CD301b⁺ DCs but are often required for

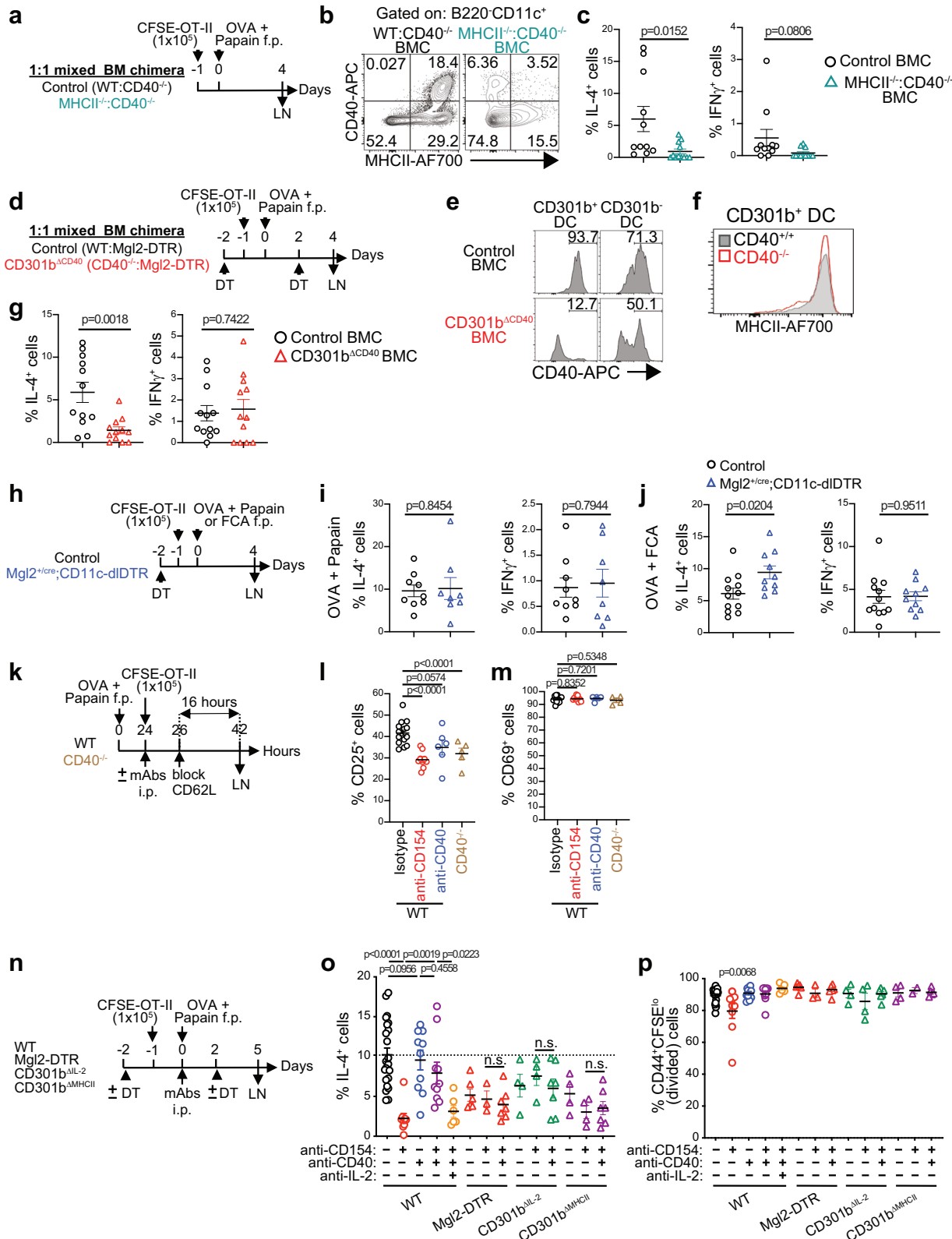

non-Th2 Th cell differentiation[3,4,91,92], under non-Th2 immunization conditions, the Th2-biased status imprinted early after priming by CD301b+ DCs may be overridden by the non-Th2 fate through the subsequent interaction with other DCs, as we recently discussed elsewhere[93]. The nature of such staggered interaction with multiple DC subsets and the role of CD301b+ DCs in these conditions need further

investigation, as the non-Th2 fate instruction cues may be delivered through antigen-independent interactions[90,91].

Taken together, our data demonstrate that CD301b+ DC-intrinsic CD40–IL-2 axis drives Th2 cell fate instruction by quantitatively regulating the IL-2R signaling in antigen-specific CD4+ T cells. While the use of a simplistic immunization approach in this study allows precise

**Fig. 6 | CD301b⁺ DCs instruct Th2 cell fate on CD4⁺ T cells through the DC-intrinsic CD40–IL-2 axis.** a–c Control (WT;CD40⁻/⁻) and MHCII⁻/⁻;CD40⁻/⁻ mixed BMC mice were transferred with OT-II cells and immunized one day later with OVA plus papain in the footpad. The dLNs were harvested 4 days after the immunization and restimulated ex vivo with PMA and ionomycin (**a**). Representative flow cytometry plots of the expression of MHCII and CD40 in B220⁻ CD11c⁺ cells (**b**) and frequencies of IL-4⁺ and IFNγ⁺ cells among the donor OT-II cells in the dLN (**c**) are shown. **d–g** As in (**a**), but control and CD301b^ΔCD40 mixed BMC mice were used as recipients (**d**). The mice were treated with DT on days −2 and +2. Representative flow cytometry histograms of CD40 in CD301b⁺ and CD301b⁻ DCs of control and CD301b^ΔCD40 mice (**e**), MHCII expression in CD301b⁺ DCs of LNs from WT and CD40⁻/⁻ mice (**f**), frequencies of IL-4⁺ and IFNγ⁺ cells among the donor OT-II cells in the dLN (**g**) are shown. **h–j** DT-treated control (*Mgl2⁺/cre*) and Mgl2⁺/cre;CD11c-dlDTR mice were transferred with OT-II cells and immunized one day later with OVA plus papain or OVA plus FCA in the footpad on Day 0. The dLNs were harvested on day 4 and restimulated ex vivo with PMA and ionomycin (**h**). Frequencies of IL-4⁺ and IFNγ⁺ cells among the donor OT-II cells in the dLN of mice immunized with OVA plus papain (**i**) and with OVA plus FCA (**j**) are shown. **k–m** OT-II cells were transferred into WT or CD40⁻/⁻ mice 24 h after immunization with OVA plus papain in the

footpad. WT mice were i.p. injected with indicated mAbs at the time of OT-II transfer. Two hours later, further homing of lymphocytes to LNs was blocked by injecting anti-CD62L mAb. The dLNs were harvested 16 h after the CD62L blockade (**k**). Frequencies of CD25⁺ (**l**) and CD69⁺ (**m**) cells among the donor OT-II cells in the dLN are shown. **n–p** WT, Mgl2-DTR, CD301b^ΔIL-2, and CD301b^ΔMHCII mice were transferred with OT-II cells and immunized one day later with OVA plus papain in the footpad and simultaneously injected i.p. with indicated mAbs. Mgl2-DTR mice were treated with DT on days −2 and +2. The dLNs were harvested 5 days after the immunization and restimulated ex vivo with PMA and ionomycin (**n**). Frequencies of IL-4⁺ cells (**o**) and CFSE^lo CD44⁺ cells (**p**) among the donor OT-II cells are shown. Data represent means ± SEM of control (*n* = 11) and MHCII⁻/⁻;CD40⁻/⁻ mixed BMC (*n* = 12) for (**c**), control (*n* = 12) and CD301b^ΔCD40 mixed BMC (*n* = 12) for (**g**), control (*n* = 9) and Mgl2⁺/cre;CD11c-dlDTR (*n* = 8) for (**i**), control (*n* = 12) and Mgl2⁺/cre;CD11c-dlDTR (*n* = 10) for (**j**), WT treated with isotype (*n* = 14), anti-CD154 (*n* = 9), or anti-CD40 (*n* = 6) mAbs, and CD40⁻/⁻ mice (*n* = 5) for (**l, m**), and 3 to 22 mice per each group for (**o, p**), or show representative flow cytometry plots of at least two independent experiments (**b, e, f**). Statistical analyses were performed using a two-tailed Student's *t*-test (**c, g, i, j, l, m, o, p**). Source data are provided as a Source Data file.

analysis of the T cell priming kinetics, the mechanism for disease-associated Th2 cell differentiation may need to be further examined in a more complex setting such as allergies and helminth infection, as such condition can potentially induce *Il4ra* even in naïve CD4⁺ T cells in an antigen-nonspecific manner[94]. Thus, whether the excess amount of type 2 cytokines released from innate immune cells under these conditions overrides the requirement of DC-derived IL-2 at the time of priming needs to be addressed in future studies.

## Methods

### Mice
Mgl2-DTR (RRID:IMSR_JAX:023822) and Mgl2-Cre (RRID:IMSR_JAX:037 286) mice were a gift from Akiko Iwasaki (Yale University). *Il2^flox/flox* mice[50] were provided by Weizhou Zhang (University of Florida). OT-II mice on the IL-4-GFP background (4get;OT-II) were a gift from Jason Weinstein (Rutgers New Jersey Medical School). Generation of Mgl2-DTR (*Mgl2⁺/DTR-EGFP*)[13], Mgl2-Cre and CD11c-dlDTR mice[20] were previously described and maintained in our colony. C57BL/6N (B6, Strain Code:027) and congenic CD45.1 (B6.SJL-*Ptprc^aPepc^b*/BoyCrCrl, Strain Code:564) mice on B6 background were purchased from Charles River Laboratory and propagated in our colony. *H2-Ab1* (MHCII)^flox/flox (B6.129×1-*H2-Ab1^b-tmKoni*/J, RRID:IMSR_JAX:013181), *Il2ra^flox/flox* (B6(129S4)-*Il2ra^tm1c(EUCOMM)Wtsi*/TrmaJ, RRID:IMSR_JAX:033093), CD40⁻/⁻ (B6.129P2-*Cd40^tm1Kik*/J, RRID:IMSR_JAX:002928), MHCII⁻/⁻ (B6.129S2-*H2^dlAb1-Ea*/J, RRID:IMSR_JAX:003584), OT-II (B6.Cg-Tg(TcraTcrb)425Cbn/J, RRID:IMSR_JAX:004194), CD207-DTR (B6.129S2-*Cd207^tm3.1(HBEGF/EGFP)Mal*/J, RRID:IMSR_JAX:016940), *Rag1⁻/⁻* (B6.129S7-*Rag1^tm1Mom*/J, RRID:IMSR_JAX:002216), Il2ra⁻/⁻ (B6;129S4-*Il2ra^tm1Dw*/J, RRID:IMSR_JAX:002462), Rosa26 loxP-stop-loxP (LSL)-tdTomato (Ai14, R26^LSL-iTom) reporter (B6.Cg-*Gt(ROSA) 26Sor^tm14(CAG-tdTomato)Hze*/J, RRID:IMSR_JAX:007914), and Nur77-GFP reporter (C57BL/6-Tg(Nr4a1-EGFP/cre)820Khog/J, RRID:IMSR_JAX:016617) mice were obtained from the Jackson Laboratory (Bar Harbor, ME) and maintained in-house.

*Mgl2^cre/cre* mice were crossed with *H2-Ab1*^flox/flox, *Il2^flox/flox*, and Il2ra^flox/flox mice to generate *Mgl2⁺/cre;H2-Ab1*^flox/flox (CD301b^ΔMHCII), *Mgl2⁺/cre;Il2^flox/flox* (CD301b^ΔIL-2), and *Mgl2⁺/cre*;Il2ra^flox/flox (CD301b^ΔCD25) mice, respectively. The leaky expression of Cre in the *Mgl2⁺/cre* mice was monitored by flow-cytometry or by qPCR and those with global deletion of MHCII, CD25, or IL-2 were excluded from the analysis as previously described[20]. *Mgl2^cre/cre* mice were also crossed with R26^LSL-iTom reporter to generate *Mgl2⁺/Cre;R26^LSL-iTom* mice and used for evaluating the Mgl2-Cre expression. CD11c (*Itgax*)-dlDTR mice were crossed with Mgl2-Cre mice to generate *Mgl2⁺/Cre;Itgax^+/dlDTR* mice. The *Mgl2⁺/Cre; Itgax^+/dlDTR* mice were further crossed with R26^LSL-iTom reporter mice to produce *Mgl2⁺/Cre;Itgax^+/dlDTR;Rosa26^+/LSL-iTom* mice. OT-II and 4get;OT-II

mice were bred to CD45.1 mice to produce CD45.1;OT-II and CD45.1;4get;OT-II mice, respectively. Nur77-GFP mice were crossed with CD45.1;OT-II mice to generate CD45.1;Nur77-GFP;OT-II mice. *Il2ra⁻/⁻* mice were crossed with CD45.1;OT-II mice to generate CD45.1;*Il2ra⁺/⁻*;OT-II and CD45.1;*Il2ra⁻/⁻*;OT-II mice. The CD45.1;*Il2ra⁺/⁻*; OT-II and CD45.1;*Il2ra⁻/⁻*;OT-II mice were maintained on the *Rag1⁻/⁻* background and further crossed onto the Nur77-GFP background where indicated. All mice were maintained in a specific pathogen-free facility at Rutgers New Jersey Medical School. For all strains, both male and female mice between 6 and 12 weeks old were used. Age and sex-matched mice were allocated randomly in the experimental groups for comparison. Control mice were co-housed or kept separately in the same room. Mice were euthanized with carbon dioxide ($CO_2$) inhalation followed by cervical dislocation. All animal experiments in this study have been approved by the Institutional Animal Care and Use Committee at Rutgers University (Protocol Number: PROTO999901093).

### BM reconstitution
For generating CD40⁻/⁻;Mgl2-DTR (CD301b^ΔCD40 BMC) and CD40⁻/⁻; MHCII⁻/⁻ mixed BMC mice, lethally irradiated (1100 cGy) B6 mice were reconstituted with a 50:50 mixture of CD40⁻/⁻ and Mgl2-DTR BM cells, or with CD40⁻/⁻ and MHCII⁻/⁻ BM cells (2 × 10⁶ cells each), respectively. Those reconstituted with WT and CD40⁻/⁻ BM cells were used as a control for CD301b^ΔCD40 and CD40⁻/⁻;MHCII⁻/⁻ mixed BMC mice. BMC mice were maintained on antibiotics (sulfamethoxazole and trimethoprim, NDC 0121-0854-16, PAI pharma) in their drinking water for 2 weeks after the reconstitution and rested for at least 6–8 weeks before being used for experiments.

### Immunization, *Nb* infection and in vivo treatment
All immunization procedures were conducted in the rear footpad with 20 μL injection volume per footpad. Mice were immunized as indicated in each Figure with 5 μg low-endotoxin OVA (Worthington Biochemical Corporation) mixed with either 50 μg papain (P4762, Sigma), 10 μL FCA (F5881, Sigma), 10 μL alum (77161, Thermo Scientific), or 10 μg CpG2216 (Invivogen) in phosphate-buffered saline (PBS). For *Nb* infection, mice were injected subcutaneously in the right ear with 600 third-stage (L3) larvae (gift from William Gause, Rutgers New Jersey Medical School) in 0.2 mL of PBS. For DC depletion, mice were injected i.p. with 500 ng DT in PBS (List Biological Laboratories) at indicated time-points. For CD40 stimulation of DCs in vivo, mice were injected i.p. with 100 μg agonistic anti-CD40 mAb (FGK4.5, BioXcell) in PBS. LNs were harvested 24 h later and restimulated with PMA and ionomycin for intracellularly staining IL-2. For mAb treatment of mice immunized with OVA plus papain, mice were injected i.p. with anti-IL-2 (S4B6-1,

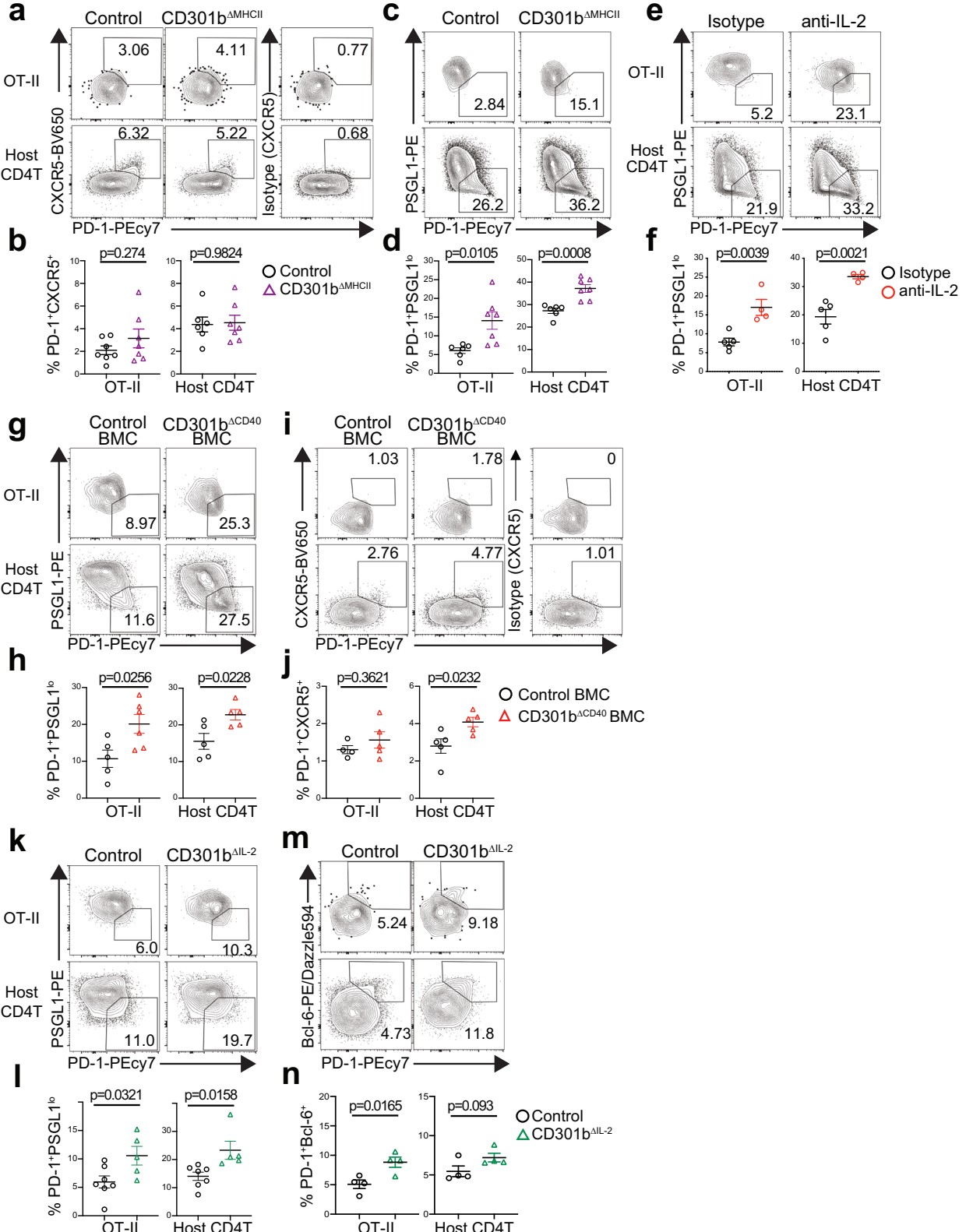

**Fig. 7 | CD301b⁺ DC-intrinsic CD40–IL-2 axis skews CD4⁺ T cells toward non-Tfh effector fate.** CD301b$^{\Delta MHCII}$ (**a-d**), WT (**e**, **f**), CD301b$^{\Delta CD40}$ BMC (**g–j**), CD301b$^{\Delta IL-2}$ (**k–n**), and respective control mice were transferred with $1 \times 10^5$ OT-II cells and immunized one day later with OVA plus papain in the footpad. In (**e**, **f**), WT mice were i.p. injected with anti-IL-2 neutralizing mAb S4B6-1 or isotype control at the time of OT-II cell transfer. The dLNs were harvested on day 7 (CD301b$^{\Delta MHCII}$, WT, and CD301b$^{\Delta CD40}$) or day 5 (CD301b$^{\Delta IL-2}$) after the immunization. Flow cytometry plots of PD-1 and CXCR5 (**a** and **i**), PD-1 and PSGL1 (**c**, **e**, **g**, **k**), or PD-1 and Bcl-6 (**k** and **m**) and frequencies of CXCR5⁺ PD-1⁺ cells (**b**, **j**), PSGL1$^{lo}$ PD-1⁺ cells (**d**, **f**, **h**, **l**) or PD-1⁺Bcl-6⁺

cells (**l** and **n**) among the donor CD44⁺ OT-II cells or host CD44⁺ CD4⁺ T cells are shown. Data represent means ± SEM of control ($n = 6$) and CD301b$^{\Delta MHCII}$ ($n = 7$) for (**b**, **d**), WT mice treated with Isotype ($n = 5$) or anti-IL-2 mAb ($n = 4$) for (**f**), control ($n = 5$) and CD301b$^{\Delta CD40}$ BMC ($n = 5$) for (**h**, **j**), control ($n = 7$) and CD301b$^{\Delta IL-2}$ ($n = 5$) for (**l**), control ($n = 4$) and CD301b$^{\Delta IL-2}$ ($n = 4$) for (**n**), or show representative flow cytometry plots of at least two independent experiments (**a**, **c**, **e**, **g**, **i**, **k**, **m**). Statistical analyses were performed using a two-tailed Student's t-test (**b**, **d**, **f**, **h**, **j**, **l**, **n**). Source data are provided as a Source Data file.

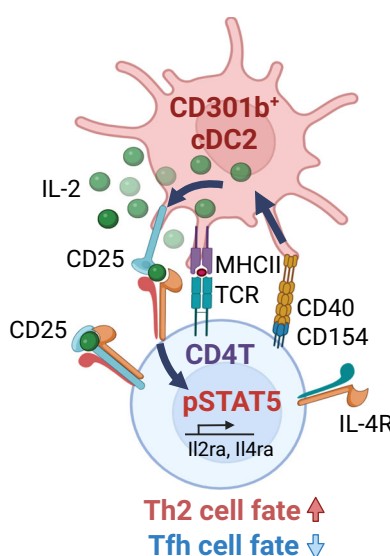

**Fig. 8 | Model for Th cell fate instruction by CD301b⁺ DCs.** Upon cognate interaction with antigen-specific CD4⁺ T cells, CD40 ligation in CD301b⁺ DCs enhances DC-intrinsic IL-2 production, which is required for the maximal CD25 expression on the cognate CD4⁺ T cells. The maximal IL-2R signaling in CD4⁺ T cells is specifically required for the Th2 fate decision and for skewing them away from the Tfh fate. Furthermore, CD301b⁺ DC-intrinsic CD25 facilitates the directed action of IL-2 toward the cognate CD4⁺ T cells. Image created with Biorender.com. (https://BioRender.com/x45b170).

BioXcell, 100 µg), anti-CD154 (MR1, BioXcell, 100 µg), anti-CD40 (FGK4.5, BioXcell, 100 µg), anti-CD154 plus anti-CD40 (100 µg each), or anti-IL-2 plus anti-CD154 plus anti-CD40 (100 µg each) mAbs at the time of OT-II transfer or immunization as indicated. Rat IgG2a isotype control (2A3, BioXcell) mAb was injected to make the number of mAbs equal across different groups.

## T cell priming assays
For adoptively transferring OT-II cells, CD4⁺ T cells were isolated from the spleen and LNs of indicated naïve OT-II strains with Mouse CD4⁺ T Cell Isolation Kit (STEMCELL Technologies 19852 or BioLegend 480033) and labeled with 1.0 µM CFSE (Thermo Fisher) or CTV (C34571, Thermo Fisher) according to the manufacturer's protocol. The purity of isolated OT-II cells was typically 90–95%. For analyzing the CD4⁺ T cell activation kinetics, CFSE- or CTV-labeled CD45.1;OT-II cells (1–2 × 10⁶) were adoptively transferred into CD45.2/.2 recipient mice that had been immunized in the rear right footpad 24 h prior. Mice with DTR expression were treated with DT as indicated in each Figure. In some experiments, multiple types of donor cells (1 × 10⁶ cells each) were co-transferred into one mouse. Two hours after the OT-II cell transfer, further LN entry of T cells was blocked by retro-orbitally injecting 100 µg anti-CD62L mAb (clone Mel-14, BE0021, BioXCell). Alternatively, in some experiments, the labeled OT-II cells were adoptively transferred into recipient mice 24 h prior to immunization and further LN entry of T cells was blocked by injecting anti-CD62L mAb at the time of immunization. The left (ndLN) and right (dLN) popliteal LNs were harvested at indicated time-points. For analyzing CD4⁺ T cell differentiation, CFSE- or CTV-labeled OT-II cells (1 × 10⁵ cells) were adoptively transferred into indicated CD45.2/.2 recipient mice, followed by immunization with OVA and an indicated adjuvant in the footpad. The dLNs were harvested at indicated time-points for intracellularly staining cytokines or intranuclear staining of Foxp3.

## Cell preparations and flow cytometry
For preparing single-cell suspensions, LNs were minced and enzymatically digested with 2.5 mg/mL collagenase D (11088882001, Sigma) in

complete RPMI-1640 medium with 10% heat-inactivated fetal bovine serum (FBS) at 37 °C for 30 min. The cells were washed with PBS containing 2 mM EDTA and then stained for dead cells with cell viability dye (Zombie Aqua or Zombie UV, BioLegend) in PBS on ice for 20 min. The cells were washed with 2 mM EDTA/PBS, incubated with 10 µg/mL anti-CD16/CD32 (2.4G2, BioLegend) on ice for 10 min to block non-specific antibody binding, and stained with fluorochrome-conjugated mAbs on ice for 20 min. For intracellular cytokine staining, LN single cell suspensions were stimulated in a 96-well round-bottom plate with Cell Stimulation Cocktail containing PMA and ionomycin (eBioscience 00-4970-03, Thermo Fisher or 423302, BioLegend) at 37 °C for 1 h, and then incubated for another 5 h at 37 °C with additional Protein Transport Inhibitor Cocktail containing Brefeldin A and Monensin (eBioscience 00-4980-03, Thermo Fisher). Cells were then fixed and permeabilized with BD Cytofix/Cytoperm Kit (BD Biosciences) and incubated with anti-cytokine mAbs for 30 min on ice. For intranuclear Foxp3 staining, cells were fixed and permeabilized with Foxp3 Transcription Factor Fixation/Permeabilization Buffer (eBioscience 00-5521-00) on ice for 30 min and incubated with anti-Foxp3 mAb on ice for 30 min. For staining phosphorylated STAT5 and GATA-3, LNs were fixed with BD Phosflow™ Fix Buffer I (BD Biosciences) at 37 °C for 10 min immediately after the harvest or ex vivo stimulation with IL-2 (50 ng/mL) or GM-CSF (50 ng/mL), and then permeabilized with BD Phosflow™ Perm Buffer III (BD Biosciences) on ice for 30 min. After blocking with anti-CD16/CD32 (2.4G2, BioLegend), cells were stained for cell surface molecules on ice for 20 min, washed with PBS containing 2 mM EDTA, and then incubated with anti-pSTAT5 and anti-GATA-3 mAbs at room temperature (20–25 °C) for 16 h.

The mAbs to CD4 (RM4-5 or GK1.5), CD8α (53-6.7), CD45R/B220 (RA3-6B2), TCRβ (H57-597), Ly6G (1A8), CD69 (H1.2F3), CD25 (PC61), CD44 (IM7), CD279 (PD-1) (RMP1-30), I-A/I-E (MHCII) (M5/114.15.2), CD11b (M1/70), CD11c (N418), CD326 (G8.8), CD301b (URA1), CD172a (P84), CD273 (PD-L2) (TY25), XCR1 (ZET), CD103 (2E7), CD40 (3/23), CD80 (16-10A1), CD86 (GL-1), CD45.1 (A20), CD45.2 (104), IFNγ (XMG1.2), IL-4 (11B11), IL-2 (JES6-5H4), GATA-3 (16E10A23), Bcl-6 (7D1), and Foxp3 (MF-14) were purchased from BioLegend. The mAb to pSTAT5 (Tyr694) (SRBCZX) was purchased from eBioscience. Anti-CXCR5 (2G8) and PSGL1 (2PH1) mAbs were purchased from BD Biosciences. All mAbs used for flow cytometry were prepared in FACS buffer (1% BSA, 2 mM EDTA, 0.05% sodium azide in PBS), except that mAbs to pSTAT5 and GATA-3 were prepared in 1x BD Perm/Wash buffer (BD Biosciences).

All samples were collected on BD LSRII (BD Biosciences) or Attune NxT (Thermo Fisher) flow cytometer and analyzed with FlowJo software (Version 9.3.2 and 10.5.0, BD).

## CITEseq data acquisition and analysis
For CITEseq, B6 mice were immunized with 5 µg OVA plus 50 ug papain in both rear footpads in 20 µL PBS. Popliteal lymph nodes were harvested from immunized mice 1 day later and digested with collagenase D (2.5 mg/mL, Roche) in RPMI-1640 medium supplemented with 10% FBS for 30 min at 37 °C, after which EDTA (5 mM final conc.) was added and incubated for another 10 min to allow complete dissociation. Pooled skin-dLNs from naive B6 mice were used as a control. Single-cell suspensions were collected, and cells were stained with viability dye (Zombie Aqua, BioBegend) in PBS for 20 min on ice. Cells were washed and incubated with anti-CD16/32 (10 µg/mL, clone 2.4G2, BioLegend) for 20 min on ice and then stained with fluorochrome-labeled mAbs for B220 and MHCII and DNA-barcoded TotalSeq mAbs for MHCII, CD11c, CD301b, CD8α, CD11b, CD64, CD103, CD172a, Ly6C and XCR1 (BioLegend) in PBS containing 1%BSA and 2 mM EDTA for 20 min on ice. The TotalSeq mAbs for MHCII, CD11c and CD301b (0.25 mg/mL) were pre-mixed with AlexaFluor 488-, AlexaFluor 546, and AlexaFluor 647-labeled antisense oligonucleotide (1 µM) that is specific to each DNA barcode, respectively, for 15 min at room temperature (20–25 °C)

in order to detect these antibodies by flow cytometry for sorting. Live MHCII$^{hi}$ CD11c$^+$ B220$^-$ cells were sorted by FACS Aria III cell sorter (BD) and used to generate Gel Beads-in-Emulsion (GEMs) using the Chromium Single-Cell 3′ Reagent v3 kit (1000092, 10X Genomics) according to the manufacturer's protocol. Briefly, cell suspensions with a > 90% viability were mixed with reverse transcription reagents and loaded onto Chromium Chip B (1000074, 10X Genomics) along with Gel beads and Partitioning oil in the recommended order, and then the chip was processed through 10X Chromium Controller for the generation of GEMs, followed by reverse transcription, cleanup, and cDNA amplification. After the cDNA amplification, size selection was used to separate the cDNA molecules for the 3′ Gexp (Gene Expression) library and ADT library construction. Library quantity and quality control were performed on Qubit 4 Fluorometer using an HS reagent kit (Q33231, Invitrogen) and TapeStation using HS DNA D1000 screen tape (5067-5584, Agilent Technologies). ADT and 3′ Gexp libraries were mixed at the ratio of 1:5 and sequenced on NovaSeq 6000 sequencer (Illumina) with a configuration of 28/8/0/91-bp for cell barcode, sample barcode and mRNA reads, respectively, as recommended by 10X Genomics. Cell Ranger and Loupe browsers were used for data analysis. The aligned data were further analyzed and visualized using Partek Flow (Partek Inc.) and Ingenuity Pathway Analysis (QIAGEN) software.

## Statistical analysis
P-values were calculated by two-tailed Student's unpaired *t*-test or by paired *t*-test as indicated using Prism software (version 8, GraphPad). Differences were defined as statistically significant when *P*-values were <0.05. Data are presented as mean ± SEM. The sample sizes were determined based on the magnitude of the effect being observed, the laboratory's previous experience, and published literature. The sample size for each experiment is provided in the figure legends. No data were excluded from the analyses. The investigators were not blinded during in vivo treatment or data collection.

## Reporting summary
Further information on research design is available in the Nature Portfolio Reporting Summary linked to this article.

## Data availability
The CITEseq data used in this study has been deposited in GEO database: accession number GSE281470. Source Data are provided with this paper. Mgl2-DTR (RRID:IMSR_JAX:023822) and Mgl2-Cre (RRID:IMSR_JAX:037286) mice have been deposited in the Jackson Laboratory. CD11-dlDTR mice are available upon request from Y.K. under an MTA from Rutgers University. Source data are provided with this paper.

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

## Acknowledgements

We thank Kendall A. Smith for sharing the *Il2^flox/flox^* mice. We thank Weizhou Zhang and Akiko Iwasaki as the original breeders of the *Il2^flox/flox^* mice and Mgl2-Cre and Mgl2-DTR mice, respectively. We thank George Yap for critical reading of the manuscript. We thank the Genomics Center, Comparative Medicine Resources, and the Flow Cytometry Core at Rutgers New Jersey Medical School for technical assistance. This work was supported by NIH grants R01AI132576, R01AI165622, and R21CA259541 to Y.K.

## Author contributions

N.T. and Y.K. designed experiments. N.T. performed most of the experiments with help from J.E.-F. and A.D.-P. J.E.-F. and Y.K. performed the CITEseq experiments and analyzed the data with assistance from the Flow Cytometry Core and the Genomics Center at Rutgers New Jersey Medical School. A.D.-P. performed the *Nb* infection experiments and analyzed the data. N.T., J.E.-F., and Y.K. performed statistical and computational analyses and interpreted the data. Y.K. conceptualized and supervised the study and acquired funding. N.T. and Y.K. wrote the paper.

## Competing interests

The authors declare no competing interests.
