## [Peer Review file · Nature Communications]

CD301b⁺ dendritic cell-derived IL-2 dictates CD4⁺ T helper cell differentiation

Corresponding Author: Dr Yosuke Kumamoto

Version 0:

Reviewer comments:

Reviewer #1

(Remarks to the Author)

In this manuscript, Tatsumi and colleagues bring elegant experimental evidence that the CD301b⁺ DC-intrinsic CD40-IL-2 axis drives Th2 cell fate instruction by quantitatively regulating the IL-2R signaling in the antigen-specific CD4 T cells. The experiments are mostly well-planned, and the data generally support the conclusions. However, several significant caveats must be addressed to increase rigor and translatability:

- The authors must perform a time course experiment (~two weeks), at least with the Mgl2-DTR and with mice in which CD301b⁺ DC lack MHC-II, and show that their findings are not limited to the time frame indicated, and this is not only a simple delay in the acquisition of IL-4-producing phenotype.
- All the experiments were performed with transgenic OT-II cells with high precursor frequency (range from 10⁵ to 2x10⁶). It is well-documented that Th phenotype depends on antigen/adjuvant dose and T cell precursor frequency. A high number of OT-II cells can also create an artificially IL-2-rich environment that would not be present in the endogenous responses. Thus, experiments should be conducted to confirm that their findings are translatable to endogenous T-cell responses and are not specific to the antigen/adjuvant dose used.
- For the activation experiments, immunizing first and then transferring the OT-II cells is very artificial. Transferring T cells into an "inflamed" animal and the LNs undergoing remodeling from the inflammation is not physiological. In physiological settings during the inflammation/immunization/infections, the T cells are already in the secondary lymphoid organs and/or recirculating. Thus, no matter the reason, the OT-II cells must be transferred before the immunization, or at the very least, shown in side-by-side experiments (matched cell numbers) that the pre-existing inflammatory environment does not affect the outcome.
- Some of the data shown here do not match those already published by the same group. For example, this paper starts with the observation that OT-II cells in mice in which CD301b⁺ DCs lack MHC-II have slightly increased total cell number, defect in IL-4 production, but intact IFN γ . Their previous Science Immunology publication showed a decrease in cell number, no significant defect in IL-4 production, but an increase in IFN γ (Fig. 7). In their original Immunity paper, the OT-II cells primed in CD301b⁺ DC-depleted hosts were severely impaired in IL-4, and the IFN γ increased. If the cognate interaction is critical in initiating the Th2 differentiation, then the data is expected to mostly match the one generated with the depletion of CD301b⁺ cells. Other discrepancies that need to be addressed. 1). The upregulation of CD69 on OT-II cells in the absence of CD301b⁺ DCs was delayed slightly, but nothing like what was previously reported by the same group. 2). Previously, in contrast to the findings presented here, the group reported that CD301b⁺ DCs alone are insufficient for generating Th2 cell responses.
- Fig. 4 shows that more CD301b⁺ DCs produce IL-2 than CD301b⁻ DCs, but in Fig. S2, on the representative flow plot, more IL-2 is associated with the CD301b⁻ DCs. Also, the percentage of CD301b⁺ seems to decrease in the absence of IL-2. Is there an autocrine IL-2 loop that regulates CD301b⁺ DC numbers? Fewer CD301b⁺ DC could lead to lower overall Th2 responses. Also, only a fraction of CD301b⁺ cells express CD25, which decreases in the Cre to the level of CD301b⁻, but it is incomplete. Does the number of CD301b⁺ DC decreases in the Cre⁺ mice?
- Are the findings presented here meaningful in any way? Please show that the observed defect matters in some way.

Minor caveats:

- In the text for Fig. 7I, J, the authors discuss mixed BMC, but the figures and figure legends do not reflect that.
- Why not use the IL-4 reporter mice throughout the experiments?
- Determining whether the defect is limited to IL-4 or other Th2 cytokines are also affected would strengthen the findings.

Reviewer #2

(Remarks to the Author)

I thought that the conclusion that CD301b+ DCs drive Th2 formation by direct antigen-presentation, IL-2 production, and provision of CD40 ligand was supported by the data. I think this work adds to be the body of knowledge about in vivo Th2 differentiation.

I have these criticism:

Drawing conclusions about the effect of IL-2 blockade or depletion of CD301b+ DCs on Th1 differentiation in immune responses (for example OVA + papain, OVA + alum, OVA + FCA) that induce very few Th1 cells is probably ill-advised. It is well established that IL-2R signaling is critical for Th1 differentiation in response to acute viral and bacterial infections PMID: 28948612, PMID: 26410627, PMID: 22271576, PMID: 22018468). I thought the Th1 experiments were a distraction and would advise removing them. This would improve the readability of the paper and focus it on the strong Th2 data. If the authors want to make statements about Th1 cells, then it would be wise to study an immune response that induces potent type 1 immunity.

GATA3 is very poorly resolved in all cases. Basing conclusions on this weak signal is probably ill advised.

The statement "these results indicate that CD301b+ DCs promote Th2 cell differentiation through reinforcing the availability of IL-2 for cognate CD4 T cells by the DC-intrinsic expression of CD25" does not make sense to this reviewer.

It was not obvious to this reviewer that the scRNA sequencing results add much to the story. The various CD40 ablation or blockade experiments are much stronger evidence that "CD301b+ DC-intrinsic CD40 is required for Th2 cell differentiation".

The Tfh marker CXCR5 is barely detected in Fig. 7A. The PD-1hi PSGL-1lo phenotype is not a standard way to identify Tfh cells. Absent evidence that these cells disappear in a Bcl-6-deficient animal it is not clear that Tfh cells are being detected. In my opinion, the paper would be stronger without this section.

Reviewer #3

(Remarks to the Author)

The manuscript of Tatsumi et al seeks to identify the dendritic cell-derived signals that program Th2 activation and differentiation. This is an important but still unanswered question and this manuscripts offers original and interesting information on this topic .

The authors use an adoptive OT-II transfer model together with targeted genetic modification of CD301b+ DC in skin lymph node to dissect the early interaction between OT-II cells and DCs, focusing on the role of IL-2, CD25 and CD40 signaling in this process. They conclude that CD40-dependent IL-2 production by CD301b+ DC, and their expression of CD25, are necessary for Th2 differentiation but not for CD4+ T cell proliferation or Th1 differentiation. While a role for IL-2 signaling in Th2 differentiation has been reported (eg ref 77 in manuscript) and is well accepted, this manuscript proposes a completely new mechanism by showing that DCs, and specifically CD301b+ DCs, are an important source of such IL-2.

The manuscript uses many sophisticated conditional KO models, and carefully timed characterizations of the lymph node immune response. While some of the conclusions are well supported by experimental evidence, others were less clear or not as convincingly demonstrated.

Major points:

- 1) The gating of CD301b+ and CD301b- populations should follow standard DC gating strategies, e.g. separating DC into migratory vs LN resident, and then into DC1, DC2 and LC as in PMID: 20038600. This gating strategy should then be kept consistent throughout the study. Instead, gating of CD301b+ and CD301b- populations is inconsistent across experiments, and especially so for the CD301b-negative population which appears to include LC, DC1 and resident DC2 populations in 4B, but only DC2s in S3H. Without a rigorous DC gating, evaluating the impact of the conditional deletions becomes impossible.
- 2) Could the authors please clarify whether the CD301b+ DC2s represent a separate subset within the migDC2s lineage, or simply a maturation stage (with all migDC2s expressing the CD301b marker at some stage of their maturation), providing references if available. This is important for understanding the results. I could not find this information in previous publications from the group.
- 3) The authors state (in title and conclusions) that the CD40-IL-2-CD25 axis is not involved in Th1 priming. However, the IFN γ responses to OVA-papain (which is used throughout the study) is only about 1-2% of OT-II, compared to 8-10 % of OT-II expressing IL-4. While FCA gave better IFN γ responses in some exps (Fig3V), this was not always the case (eg Fig S1C).

Therefore, the effect on Th1 could not be evaluated because little to no IFN γ was made after immunization. A robust protocol of Th1 induction is needed.

- 4) IL-2 expression in CD301b⁺ DCs is a key step of the proposed mechanism of Th2 priming. However, IL-2 expression in DCs is shown only after PMA+Iono stimulation, and is very low in the CD301b⁺ subset. The authors need to provide evidence of IL-2 protein expression by DC *in vivo*, or alternatively by anti-CD40-stimulated DCs *ex vivo*. It would be interesting to see if the Il2 gene can be resolved in the scRNAseq dataset shown in figure 5.
- 5) The manuscript uses many elegant models of conditional gene KO in CD301b⁺ DCs and antibody-mediated blocking, but does not state whether these mice were assessed to determine potential impacts of mutation/treatments on eg T cell differentiation and homeostasis, numbers and ratios of conventional T cells and Tregs, etc. For example, increased IL-2 production by DCs has been associated with increased Treg frequency (Whyte et al PMID: 35699942). Was Treg frequency altered in the CD301b⁻ IL-2?
- 6) The introduction presents a model where the differentiation of various Th subsets (Th1, Th2...) is exclusively determined by CD4⁺ T cell priming in the context of different DC subsets. This does not account for other publications showing promiscuity, which is consistent with, for example, DC subsets other than DC1 being able to produce IL-12 when exposed to different stimuli (eg, PMID: 33159073 11466361). Broadening the introduction to include other models would in my view better reflect existing literature as well as provide a better starting point to approach the results.
- 7) The manuscript Figures are large and crowded (for example, figures 3 and 4 each include more than 30 panels) and could be made easier to follow and see by moving all non-essential data to Supplemental.

Minor:

- 1) An important role of T-cell intrinsic CD25 expression in Th1 and Th2 effector differentiation was demonstrated in PMID: 22018468 and 26750312, these papers should be cited.
- 2) Can the authors rule out pSTAT5 (Figs 2, 3, 4) being induced during sample preparation rather than *in vivo*? The LN digestion step (30' at 37C according to the Methods) is long enough to extinguish IL-2 signaling that had taken place *in vivo*.
- 3) Several Figures show frequencies of OT-II cells expressing specific markers, but often there are no clear positive and negative populations to be identified (Figs 2J-K, 3I...). Comparing median fluorescence intensity of the whole population would be more appropriate in these cases.
- 4) Figure 3T-U shows very different frequencies of IL-2ra^{+/+} vs. IL-2ra^{-/-} T cells in the same mouse after immunization (3T), but identical numbers of divisions/no division in the two populations (3U). How can these data be reconciled? Please discuss.
- 5) Figure 3V-W and S1F: I agree that there might be stronger inhibition in the IL-4⁺ populations, nonetheless the IFN γ populations are also clearly inhibited and this should be acknowledged. Differences in inhibition might be simply due to differences in the dose response of each cytokine.
- 6) What is the difference between the FCA experiments in 3V-W and S1G-I? The legend to 3V-W does not mention FCA but FCA is included in 3R-W.
- 7) 6P: here the response is expressed as % divided cells in the CD44⁺ population. Comparing % divided in the total OT-II population would enable a more quantitative comparison.
- 8) There is no CXCR5 staining in Figure 7. Either the staining did not work, or the time of analysis was not appropriate.

Version 1:

Reviewer comments:

Reviewer #1

(Remarks to the Author)

The authors revised the manuscript in accordance with reviewers' comments, which increased its clarity and impact.

Reviewer #3

(Remarks to the Author)

I would like to thank the authors for updating their manuscript and making several points clearer including DC gating and phenotype, a discussion of CD301b⁺ as a subset vs maturation marker, background on the cKO mouse strains used etc.

However, my major point 3 has been addressed only in part. While Figures 1 and 3 have been modified to include a model of OVA+CpG immunization which gives strong Th1 priming, and nicely confirm the Authors' conclusions, other figures including Figs 3, 4 and 6 still rely on very low IFN γ measurements after immunization in Th2 conditions using papain+OVA to conclude that findings in Th2 setting do not extend to Th1 settings. This conclusion cannot be based on OVA+papain data and require data from an OVA+CpG experiment. Papain+OVA does not elicit a Th1 response thus the relative panels are not informative and should be removed together with the related comments and discussion.

Similarly, in Figure 3j, the authors compare normalized IL-4 and IFN γ responses to each other, to conclude that "Th2 cell differentiation relies more stringently on IL-2R signaling than Th1". However, any differences in 3j are only due to low responses in 3i. Panel 3j should be removed from the Figure together with the "more stringent" claims which in my opinion are not justified.

Minor:

Fig S1A, it is surprising that all XCR1⁺ cells are also CD326⁺

The statement "reduced expansion was not due to poor survival (Fig. 3h), suggesting that it was instead due to differences in proliferation that was not detectable within the range of CFSE dilution we used" does not make sense. Data in 7A show that there are almost no TFH cells in all conditions (or the staining didn't work), this panel should be removed

Reviewer #4

(Remarks to the Author)

The authors have adequately addressed this reviewer's concerns.

Minor point: It is not stated which statistical test was used to determine the significance of the multiple comparisons in Fig. 4b

Version 2:

Reviewer comments:

Reviewer #3

(Remarks to the Author)

I would like to thank the Authors for clarifying their points. I have no further questions.

Reviewer's Comments:

Our responses are shown in blue below:

Reviewer #1 (Remarks to the Author)

In this manuscript, Tatsumi and colleagues bring elegant experimental evidence that the CD301b⁺ DC-intrinsic CD40-IL-2 axis drives Th2 cell fate instruction by quantitatively regulating the IL-2R signaling in the antigen-specific CD4 T cells. The experiments are mostly well-planned, and the data generally support the conclusions. However, several significant caveats must be addressed to increase rigor and translatability:

We appreciate the Reviewer #1 for the positive feedback and constructive criticism. We have now added more data and edited the writing to address the concerns. Please see our responses to each comment below.

- 1. The authors must perform a time course experiment (~two weeks), at least with the Mgl2-DTR and with mice in which CD301b⁺ DC lack MHC-II, and show that their findings are not limited to the time frame indicated, and this is not only a simple delay in the acquisition of IL-4-producing phenotype.*

We appreciate this reviewer for raising this important point. We would like to first clarify that the requirement of CD301b⁺DCs for Th2 cell differentiation is not limited to the synchronized priming of OT-II cells or to the papain immunization model we used in this study, as we previously showed that the depletion of CD301b⁺ DCs results in impaired Th2 cell differentiation of endogenous polyclonal CD4T cells upon infection with a helminth parasite *Nippostrongylus brasiliensis* (*Immunity* 39:733), which we now also found in our newly added data in the CD301b^{ΔMHCII} mice (New **Fig. 1l-n**). Thus, we believe that the requirement of the cognate interaction with CD301b⁺DCs for Th2 fate decision by CD4T cells is not specific to the time-points or the immunization models we use.

To further exclude the possibility that the defect in Th2 differentiation we observed simply reflects a delay in Th2 phenotype acquisition, as suggested by the reviewer, we repeated some of the experiments by collecting the dLNs on day 10 or 14 (**Supplementary Fig 2f-h**) instead of day 7 (**Fig.1a-h**). The number of OT-II cells declined from day 7 (**Fig.1c**) to day 10/ 14 as the effector T cells leave the dLN, but the frequency of IL-4⁺OT-II Th2 cells remained reduced for up to 14 days in both CD301b-depleted and CD301b^{ΔMHCII} mice compared to the control mice, indicating that the impaired acquisition of IL-4 expression in those mice is not due to delayed activation but it is rather due to a qualitative change in OT-II cell differentiation.

- 2. All the experiments were performed with transgenic OT-II cells with high precursor frequency (range from 10⁵ to 2x10⁶). It is well-documented that Th phenotype depends on antigen/adjuvant dose and T cell precursor frequency. A high number of OT-II cells can also create an artificially IL-2-rich environment that would not be present in the endogenous responses. Thus, experiments should be conducted to confirm that their findings are translatable to endogenous T-cell responses and are not specific to the antigen/adjuvant dose used.*

The high number of OT-II cells provided us with necessary resolution for the analyses shown, but we agree with the Reviewer that high frequency of antigen-specific naïve CD4T cell clones might skew the effector outcome. However, we have previously shown that depletion of CD301b⁺ DCs led to impaired IL-4 production from the *endogenous* polyclonal CD4 T cells in the skin and lung dLNs upon subcutaneous infection with *Nb* (*Immunity* 39:733), which we have

now repeated in the CD301b^{AMHClI} mice (Fig 11-n). The reduced IL-4 production from the endogenous polyclonal CD4 T cells suggests that the cognate interaction with CD301b⁺DCs is indeed required for the Th2 fate decision.

3. For the activation experiments, immunizing first and then transferring the OT-II cells is very artificial. Transferring T cells into an "inflamed" animal and the LNs undergoing remodeling from the inflammation is not physiological. In physiological settings during the inflammation/immunization/infections, the T cells are already in the secondary lymphoid organs and/or recirculating. Thus, no matter the reason, the OT-II cells must be transferred before the immunization, or at the very least, shown in side-by-side experiments (matched cell numbers) that the pre-existing inflammatory environment does not affect the outcome.

While we agree that there are certain limitations, like in many other studies, the use of an artificial system to synchronize the priming was necessary for us to dissect the activation kinetics of CD4T cells at the clonal level *in vivo*. Our rationale for transferring naïve OT-II cells into pre-immunized mice was based on the assumption that, under physiological conditions, the majority of OVA-specific naïve endogenous CD4T cells would likely enter the dLN after the immunization, because the total number of OVA-specific naïve CD4T cell clones in WT B6 mice has been estimated to be roughly 1×10^{-7} in frequency and only about 16 cells per whole body (*Immunity* 27:203), and the average total cellularity of naïve popliteal LN is about 1-2 million cells (*PNAS* 108:8749), making the average number of naïve endogenous OVA-specific CD4T clones in naïve mice less than one cell per popliteal LN at any given time-point.

It is well-documented that the inflammation-induced dLN remodeling increases the influx of circulating naïve T cells into the dLNs, leading to more efficient scanning of rare antigen-specific T cell clones by DCs and thus maximizing the magnitude of adaptive immunity (*PNAS* 102:16315; *PNAS* 108:8749). This suggests that, in addition to the relatively rare antigen-specific naïve T cell clones that resided in the dLNs already at the time of immunization, T cells that arrive in the dLNs during the LN remodeling significantly contribute to immune responses. Indeed, naïve CD4T cells that enter the dLNs after subcutaneous antigen injection participate in the primary T cell priming, accounting for approximately 50% and 90% of the population of antigen-specific CD4T cells at the peak of clonal expansion and at the contraction phase of the primary response, respectively (*J Exp Med* 203:1045). In fact, many studies have utilized synchronized priming models like ours in order to dissect the T-cell priming dynamics *in vivo*, where T cells were transferred into pre-immunized mice (*Nature* 427:154; *J Exp Med* 198:715; *J Exp Med* 200:847; *Immunity* 37:1091). In these studies, mice were first immunized subcutaneously with antigens or antigen-loaded DCs with adjuvants before the adoptive transfer of T cells, with or without blocking further T cell entry a few hours later to synchronize the priming. Thus, although transferring high numbers of OT-II cells into pre-immunized mice is indeed an artificial model, we believe that it is nonetheless useful to precisely examine the kinetics of early T cell responses.

That said, it is also true that the data from this approach may not represent the response of the rare, probably a minor, population of antigen-specific naïve CD4T cell clones that were already present in the dLN at the time of immunization. To address this issue, as suggested by this Reviewer, we now examined the activation kinetics of OT-II cells adoptively transferred prior to immunization and found a similar reduction in CD25, pSTAT5, or GATA-3 in mice depleted of CD301b⁺DCs (Supplementary Fig.2d,e). These data suggest the requirement of CD301b for the optimal IL-2R signaling in antigen-specific CD4T cells and their subsequent Th2 differentiation regardless of the timing of their encounter with the antigen.

4. Some of the data shown here do not match those already published by the same group. For example, this paper starts with the observation that OT-II cells in mice in which CD301b⁺ DCs lack MHC-II have slightly increased total cell number, defect in IL-4 production, but intact IFN γ . Their previous *Science Immunology* publication showed a decrease in cell number, no significant defect in IL-4 production, but an increase in IFN γ (Fig. 7). In their original *Immunity* paper, the OT-II cells primed in CD301b⁺ DC-depleted hosts were severely impaired in IL-4, and the IFN γ increased. If the cognate interaction is critical in initiating the Th2 differentiation, then the data is expected to mostly match the one generated with the depletion of CD301b⁺ cells. Other discrepancies that need to be addressed. 1). The upregulation of CD69 on OT-II cells in the absence of CD301b⁺ DCs was delayed slightly, but nothing like what was previously reported by the same group. 2). Previously, in contrast to the findings presented here, the group reported that CD301b⁺ DCs alone are insufficient for generating Th2 cell responses.

We thank this Reviewer for following our previous studies in depth. The reason why some readouts differ between experiments is due to two different experimental conditions (that is, the non-synchronized priming and the synchronized priming with CD62L blockade) as discussed in detail in our recent *Science Immunology* paper (*Sci Immunol* 6:eabg0336). Since this comment is mainly directed to the differences in data we reported in our previous papers, we wish to avoid reiterating the same explanation we already discussed in those papers in this manuscript, but below we summarize the key differences between different experimental conditions for reviewing purposes. Importantly, we believe that there are no discrepancies between this manuscript and our previous studies.

Role of CD301b⁺ DCs in OT-II cell expansion

As this Reviewer pointed out, the impact of CD301b⁺ DC depletion on the OT-II cell number differs by the experimental condition. Under the “synchronized” condition, the expansion of the donor OT-II cells was diminished at 56 hours after the CD62L blockade in CD301b⁺ DC-depleted mice when a high number (2×10^6) of OT-II cells were transferred (Fig. 7D,7E in *Sci Immunol* 6:eabg0336). In contrast, when the mice were transferred with titrated low numbers of OT-II cells under the “non-synchronized” condition without CD62L blockade, their priming was impaired in CD301b⁺ DC-depleted mice only when the number of input OT-II cells was 1000 cells or lower, whereas no difference in priming was observed when the mice were transferred with higher numbers (10,000 cells) of OT-II cells (Fig. 8A-8C in *Sci Immunol* 6:eabg0336). Similar data were observed in the CD301b^{AMHClI} mice (Fig. 8D in *Sci Immunol* 6:eabg0336). A likely explanation for the lack of impaired priming in CD301b⁺ DC-depleted mice with higher numbers of input OT-II cells under the “non-synchronized” model is because the lagged priming of recirculating OT-II cells can mask the impact of CD301b⁺ DC depletion on priming efficacy, as discussed in detail in that paper. Consistent with these observations, the expansion of OT-II cells was intact or slightly increased on day 7 under the “non-synchronized” conditions in the CD301b^{AMHClI} mice when 1×10^5 OT-II cells were transferred (Fig. 1c, 1g and 1j of this manuscript). While we do not fully understand where exactly this increase is coming from, we previously reported that the depletion of CD301b⁺ DCs results in a temporary reduction of CD4T cells in the dLN early after priming, followed by a quick recovery due to a greater expansion of Tfh cells (*Elife* 5:e17979). In agreement, the mild increase in OT-II cells in the CD301b^{AMHClI} mice partially reflects the increase in PD-1⁺ PSGL1^{lo} pre-Tfh-like cells upon immunization with OVA plus papain (Fig. 7c-7d of this manuscript).

Role of CD301b⁺ DCs in Th2 and Th1 differentiation

As this Reviewer pointed out, under the synchronized condition, our previous studies showed only a modest, statistically insignificant reduction in intracellular IL-4 staining in OT-II cells in CD301b⁺ DC-depleted mice (Fig. 7F and 7I in *Sci Immunol* 6:eabg0336), which was in contrast with the reduced IL-4 staining in OT-II cells primed under the non-synchronized condition (for example, Fig. 7A and 7C in *Immunity* 39:733). However, the lack of difference in IL-4 staining

seemed to be a technical issue due to poor separation in intracellular staining specifically in the synchronized condition, since, for the reasons we do not fully understand, the sensitivity of the intracellular IL-4 staining in OT-II cells in WT recipients appears to be generally lower in the synchronized model (2-3 % out of OT-II cells) than in the non-synchronized model (about 10 % out of OT-II cells). In fact, the expression of IL-4Ra and IL-4-GFP reporter was both impaired in OT-II cells in CD301b⁺DC-depleted mice even under the synchronized condition (Fig.7G and 7H in *Sci Immunol* 6:eabg0336), indicating that their Th2 differentiation was actually impaired despite the lack of sensitivity in detecting the differences in intracellular IL-4. This is also supported by the reduced GATA3 expression in OT-II cells primed under the synchronized condition in CD301b⁺ DC-depleted mice (**Supplementary Fig.S2e** of this manuscript).

Regarding Th1 differentiation, we previously showed a significant increase in IFN γ ⁺ cells in CD301b⁺ DC-depleted mice under the synchronized condition (Fig 7F and 7I in *Sci Immunol* 6:eabg0336). We also found that IFN γ production was slightly increased in CD301b⁺ DC-depleted mice following immunization with papain under the non-synchronized condition (Fig. 7B in *Immunity* 39:733) or upon infection with herpes simplex virus (Fig. S6C in *Immunity* 39:733). Consistent with these, GATA-3 expression is reduced but T-bet expression is increased in OT-II cells by day 4 in CD301b⁺DC-depleted mice following immunization with papain (Fig. 7A in *Immunity* 39:733). However, IFN γ production upon immunization with CpG was not affected by the depletion of CD301b⁺ DCs under the non-synchronized condition (Fig. S6D in *Immunity* 39:733). Similarly, there was no change in IFN γ production in the CD301b^{MHCII} mice upon immunization with OVA plus various adjuvants under the non-synchronized condition (**Fig. 1e, 1h and 1k** of this manuscript). Overall, the compensatory increase of IFN γ ⁺ cells in the absence of CD301b⁺ DCs seems to be more pronounced under the synchronized conditions than in the non-synchronized conditions (Fig. 7F and 7I in *Sci Immunol* 6:eabg0336). As discussed in our previous paper (*Sci Immunol* 6:eabg0336), we speculate that, under the non-synchronized condition, the lagged priming of recirculating OT-II cells had masked the impact of CD301b⁺ DC depletion on Th1 differentiation because the critical time window for the interaction between CD301b⁺ DCs and antigen-specific CD4 T cells is narrow for newly arrived naïve CD4T cells (Fig. 7B in *Sci Immunol* 6:eabg0336).

Sufficiency of CD301b⁺ DCs for generating Th2 cells

As this Reviewer pointed out, in our previous experiments (Fig. S7B in *Immunity* 39:733), the injection of OVA₃₂₃₋₃₃₉-pulsed CD301b⁺ MHCII^{hi} DCs sorted from papain-immunized WT donor into the footpad of naïve mice failed to induce Th2 cells, suggesting that CD301b⁺ DCs alone are not sufficient in inducing Th2 cell differentiation solely by themselves. It was possible, however, that adoptively transferring the isolated DCs into naïve hosts did not fully mimic the *in vivo* conditions, such as non-DC components that need to be activated by the adjuvant for the effective Th2 cell differentiation, even if CD301b⁺ DCs were actually sufficient as an antigen-presenting cell population. In this study, we thus reinvestigated this sufficiency question at a different level by using the Mgl2^{+/Cre};CD11c^{+/dDTR} mice, in which CD301b⁺ DCs are protected from DT-induced depletion while CD301b⁻ DCs are depleted by DT, whereas all other non-DC components should theoretically remain intact as in WT mice (**Supplementary Fig. 5h-5m** of this manuscript). Using this model, we found that CD301b⁺ DCs were indeed sufficient for inducing Th2 cells when all conditions other than the DC compartment fully mimic the WT environment (**Fig.6h-6j** of this manuscript), which was now clarified in the text.

Role of CD301b⁺ DCs in CD69 upregulation

This Reviewer pointed out that “the upregulation of CD69 on OT-II cells in the absence of CD301b⁺ DCs was delayed slightly, but nothing like what was previously reported by the same group.” In our recent paper (*Sci Immunol* 6:eabg0336), we reported that, under the synchronized condition, the CD69 expression in the donor OT-II cells in CD301b⁺DC-depleted

mice was reduced at earlier time points (54.5% in WT vs 26.6% in CD301b⁺DC-depleted mice at 0 h, 96.5% in WT vs 91.9% in CD301b⁺DC-depleted mice at 8 h after the CD62L blockade), but was later recovered at 16 h time point (96.8% in WT vs 95.2% in CD301b⁺DC-depleted mice) (Fig. 3D and Fig. 6G-H in *Sci Immunol* 6:eabg0336). Consistently, the current study reproduced the delayed CD69 upregulation under the same experimental condition (54.1% in WT and 29.3% in Mgl2-DTR at 0 h, 95.6% in WT and 89.6% in Mgl2-DTR at 8h, and 90.2% in WT and 92.0% in Mgl2-DTR at 16h, **Fig. 2b** in this manuscript). The only difference that was NOT reported in our previous studies is the impaired upregulation of CD25 during the priming process in the absence of CD301b⁺ DCs (**Fig. 2c-2i** in this manuscript), which we found critical for the Th2 fate instruction in this study.

Thus, we believe that the data presented in this manuscript are fully consistent with our previous studies.

5. *Fig. 4 shows that more CD301b⁺ DCs produce IL-2 than CD301b⁻ DCs, but in Fig. S2, on the representative flow plot, more IL-2 is associated with the CD301b⁻ DCs. Also, the percentage of CD301b⁺ seems to decrease in the absence of IL-2. Is there an autocrine IL-2 loop that regulates CD301b⁺ DC numbers? Fewer CD301b⁺ DC could lead to lower overall Th2 responses. Also, only a fraction of CD301b⁺ cells express CD25, which decreases in the Cre to the level of CD301b⁻, but it is incomplete. Does the number of CD301b⁺ DC decreases in the Cre⁺ mice?*

IL-2 expression in CD301b⁺ DCs

We apologize for the lack of clarity in our original manuscript. We have now clarified the gating strategy for the IL-2 levels in DC subsets (**Supplementary Fig. 4a and 4b** in this manuscript, please also see our response to the Major Point # 1 by the Reviewer#3) and revised the sentence to “we found that CD301b⁺ DCs in the dLN expressed significantly higher levels of IL-2 than those in the non-draining lymph node (ndLN), while it was not clearly observed in other DC subsets” to describe the data shown in **Fig. 4b and Supplementary Fig. S4b** more accurately. The data suggest that the upregulation of IL-2 upon cognate interaction or CD40 stimulation is specific to CD301b⁺ DCs, but that does not necessarily mean that CD301b⁺ DCs naturally express higher levels of IL-2 than all other DC subsets (for instance, see the left ndLN in **Fig.4b** and isotype control in **Fig.5i**, as well as the mRNA expression in **Fig.5d** and the *Mgl2^{+/Cre}* control in **Supplementary Fig.S4e** in mice immunized only with papain without OVA and OT-II cell transfer).

Possible regulation of CD301b⁺ DC numbers by autocrine IL-2

We thank this Reviewer for this interesting question. While our experiments do not formally exclude the possibility of autocrine IL-2 signaling in CD301b⁺ DCs, we believe that the possible autocrine IL-2 signaling in CD301b⁺ DCs, if any, is NOT the main cause for the phenotype we observed in the CD301b^{ΔIL-2} mice for the following reasons: First, as shown in **Fig. 4I and Supplementary Fig. S4i-k**, CD301b⁺ DCs express CD25 and CD132, but they do not seem to express CD122, which is required for IL-2 signaling. Second, to functionally validate this finding, we now show that *ex vivo* stimulation with IL-2 (50 ng/ml) induced phosphorylation of STAT5 in CD4T cells but not in either CD301b⁺ or CD301b⁻ DCs, whereas stimulation with GM-CSF (50 ng/ml) induced pSTAT5 in DCs but not in CD4 T cells (**Supplementary Fig. S4I**), indicating that DCs are generally less sensitive to IL-2 than CD4T cells. This is consistent with earlier studies that showed the lack of IL-2 signaling in DCs (*Nat Med* 17:104; *Nature* 533:110), though others reported intact IL-2 signaling in GM-CSF-induced BMDCs or cultured splenic DCs *in vitro* (*Eur J Immunol* 30:1453; *Eur J Immunol* 45:1494). Third, we found no change in the numbers of CD301b⁺ and CD301b⁻ DC subsets in the CD301b^{ΔIL-2} and CD301b^{ΔCD25} mice (**Supplementary Fig. S4f, S4o**), further ruling out the potential reduction of CD301b⁺ DCs as the main cause for

the impaired Th2 responses in those mice. Thus, the possible autocrine IL-2 signaling in CD301b⁺DCs, if any, is not the likely cause for the impaired Th2 differentiation phenotype we observed in the CD301b^{ΔIL-2} mice and CD301b^{ΔCD25} mice.

Deletion efficiency of CD25 and the number of CD301b⁺ DCs in the *Mgl2*^{+/*Cre*} mice

As to the deletion efficacy of CD25 in the CD301b^{ΔCD25} mice, while the deletion may not be complete, it does not affect the interpretation of our data since we use the *Mgl2*^{+/*Cre*} mice as the control group for all the conditional KO (CD301b^{ΔMHCI}, CD301b^{ΔIL-2}, and CD301b^{ΔCD25}) experiments. Thus, our data suggest that even a partial reduction in CD25 expression in CD301b⁺ DCs has a significant impact on Th2 cell development. As to the number of CD301b⁺ DCs in the *Mgl2*^{+/*Cre*} mice, it is technically difficult to directly compare their numbers to those in WT (*Mgl2*^{+/+}) mice, since the IRES-Cre construct inserted into the *Mgl2* 3'UTR seems to shut down the endogenous *Mgl2* expression *in cis*, thereby reducing the CD301b protein level in CD301b⁺ DCs in those mice by roughly 50% compared to that in WT mice as we previously reported (Fig. S2 in *Sci Immunol* 6:eabg0336). However, by using PD-L2 as a surrogate marker for CD301b⁺DCs as shown by others (*Immunity* 39:722), we showed previously that the number of the total PD-L2⁺DCs in the *Mgl2*^{+/*Cre*} mice was comparable to that in WT mice (Fig. S2D in *Sci Immunol* 6:eabg0336), suggesting that the presence of the *Mgl2*^{*Cre*} allele alone does not affect the development of CD301b⁺DCs.

6. *Are the findings presented here meaningful in any way? Please show that the observed defect matters in some way.*

We appreciate the Reviewer for this fundamental criticism. While we are not entirely sure what exactly the Reviewer meant by “the observed defect matters in some way”, we would like to emphasize that our manuscript is not necessarily focusing on reporting a specific defect in a single mouse model but rather it is about the detailed mechanism of the instruction of effector fate of CD4T cells by DCs. We believe that the strength of our study lies in the fact that all the genetic models we used yielded a surprisingly similar phenotype with impaired Th2 differentiation and enhanced commitment toward the pre-Tfh fate, *collectively* pointing out the importance of the CD301b⁺ DC-intrinsic CD40–IL-2 axis in the effector fate instruction of antigen-specific CD4T cells. However, to further ensure the generalizability of our findings, we have now added new data showing impaired Th2 cell differentiation of endogenous polyclonal CD4 T cells in the CD301b^{ΔMHCI} mice in response to infection with a helminth parasite *Nippostrongylus brasiliensis*, suggesting the requirement of cognate interaction with CD301b⁺ DCs for Th2 cell differentiation in this commonly used parasite infection model (Fig. 1l-n). Furthermore, our data indicate that CD301b⁺ DC-derived IL-2 regulates the bifurcation between the pre-Tfh and non-Tfh effector fates not only in OT-II cells but also in endogenous CD4T cells, again suggesting its general importance in the fate decision by the polyclonal naïve CD4T cell pool (Fig. 7). That said, it is not our intent to fully exclude the possibility that some Th2 cells may differentiate independently of the CD301b⁺ DC-intrinsic CD40–IL-2 axis, which is clearly mentioned in the last paragraph of Discussion.

Minor caveats:

7. *In the text for Fig. 7I, J, the authors discuss mixed BMC, but the figures and figure legends do not reflect that.*

CD301b^{ΔCD40} mice (Fig. 7i and 7j) are mixed BM chimera and their labels are now changed to “CD301b^{ΔCD40} BMC”. It is also clearly described in the “BM reconstitution” section in Methods as well as in Fig. 6d.

8. *Why not use the IL-4 reporter mice throughout the experiments?*

While the IL-4-GFP reporter expression is more sensitive than intracellular IL-4 staining, the GFP reporter expression does not always guarantee the production of IL-4 protein (*Nat Immunol* 7:644). We therefore stained the IL-4 protein intracellularly in most of our experiments.

9. *Determining whether the defect is limited to IL-4 or other Th2 cytokines are also affected would strengthen the findings.*

Consistent with our previous study (*Immunity* 39:733), in addition to the impaired IL-4 production, the expression of GATA3, the master regulator of Th2 cell differentiation, was also impaired in OT-II cells with reduced IL-2R signaling (**Supplementary Fig. S3l and S3m**) as well as in those primed in the CD301b⁺DC-depleted mice (**Fig. 2j and 2k, Supplementary Fig. S2e**). However, we were not able to detect IL-5 or IL-13 at reliable levels, presumably because these cytokines require additional licensing by tissue-derived cytokines such as TSLP, IL-33, and IL-25 at the peripheral tissue (*Nat Immunol* 13:58; *Nat Immunol* 17:1381). In agreement, we found higher expression of IL-5 and IL-13 in CD4T cells in the lung than in the skin- and lung-dLNs upon Nb infection in WT mice. Notably, the CD301b^{AMHClI} mice showed lower frequencies of IL-4⁺, IL-5⁺ and IL-13⁺ CD4T cells in the lung, indicating that cognate interaction with CD301b⁺ DCs is indeed required for the production of all three cytokines (see **Response Fig. 1** below). However, it is currently unclear if the expression of IL-5 and IL-13 by Th2 cells requires antigen presentation by CD301b⁺ DCs at the time of priming in the dLN or during their terminal differentiation in the lung. Since this manuscript focuses on the role of CD301b⁺ DCs specifically in CD4T cell priming in the dLN, we wish to only show the data in the dLNs focusing on IL-4 production (**Fig. 1l-n**).

[Redacted]

Reviewer #2 (Remarks to the Author):

I thought that the conclusion that CD301b⁺ DCs drive Th2 formation by direct antigen-presentation, IL-2 production, and provision of CD40 ligand was supported by the data. I think this work adds to be the body of knowledge about in vivo Th2 differentiation.

I have these criticism:

We appreciate the Reviewer #2 for insightful suggestions and constructive criticism. We have now added more data and edited the writing to address the concerns. Please see our responses to each comment below.

- 1. Drawing conclusions about the effect of IL-2 blockade or depletion of CD301b⁺ DCs on Th1 differentiation in immune responses (for example OVA + papain, OVA + alum, OVA + FCA) that induce very few Th1 cells is probably ill-advised. It is well established that IL-2R signaling is critical for Th1 differentiation in response to acute viral and bacterial infections PMID: 28948612, PMID: 26410627, PMID: 22271576, PMID: 22018468). I thought the Th1 experiments were a distraction and would advise removing them. This would improve the readability of the paper and focus it on the strong Th2 data. If the authors want to make statements about Th1 cells, then it would be wise to study an immune response that induces potent type 1 immunity.*

We appreciate this Reviewer for the insightful comment. Regarding previous studies showing the critical role of IL-2R signaling for Th1 differentiation, PMID 22271576 and 26410627 were already cited in our original manuscript (refs 71 and 72 in the revised version) and PMID 28948612 and 22018468 are now additionally cited in the revised manuscript (refs 88 and 89, respectively). We agree with the Reviewer that the models we used in our original manuscript were probably not the best models to evaluate Th1 differentiation, but we wish to keep the data in the manuscript as we believe that the comparison between Th2 and Th1 differentiation under the same immunization conditions would provide important information to the readers. To further clarify the role of CD301b⁺ DCs and IL-2R signaling in Th1 responses, we now added data from mice immunized with OVA plus CpG as described below (**Fig. 1i-1k**, **Supplementary Fig. S2a-S2c**, and **Fig. 3**). Please also see our response to the point #4 "Role of CD301b⁺ DCs in Th2 and Th1 differentiation" by the Reviewer #1.

Role of CD301b⁺ DCs in priming and Th1 differentiation of OT-II cells upon immunization with OVA plus CpG

Consistent with our previous study (Fig. S6D in *Immunity* 39:733), immunization with OVA plus CpG induced a strong Th1 response (>10% IFN γ ⁺ cells with minimal IL-4 production) in OT-II cells but the depletion of CD301b⁺ DCs did not alter the response (**Fig. 1i-k**). However, similarly to the Th2 differentiation upon OVA plus papain immunization, the upregulation of CD25 and CD69 in OT-II cells was partially impaired in CD301b⁺ DC-depleted mice (**Supplementary Fig. S2a-S2c**). These results suggest that CD301b⁺ DCs are similarly required for the timely priming and full CD25 upregulation in OT-II cells under both Th1 and Th2 conditions, but the partial reduction of CD25 expression in OT-II cells does not affect their Th1 differentiation.

Role of IL-2R signaling in CD4 T cells for Th1 differentiation

As in mice immunized with OVA plus FCA, both complete (*Il2ra*^{-/-}) and partial (*Il2ra*^{+/-}) loss of CD25 in OT-II cells reduced IFN γ production to 38.6 \pm 7.4 % and 61.6 \pm 11.1 % of WT OT-II cells, respectively, in mice immunized with OVA plus CpG (new **Fig. 3i-3j**), indicating the

requirement of IL-2R signaling for Th1 cell differentiation. However, these reductions were less significant than the reduction of Th2 cells observed in mice immunized with OVA plus papain (4.7 ± 1.1 % in *Il2ra*^{-/-} and 21.6 ± 4.17 % in *Il2ra*^{+/-} vs. WT) or FCA (17.1 ± 3.4 % in *Il2ra*^{-/-} and 9.7 ± 3.6 % in *Il2ra*^{+/-} vs. WT) (**Fig.3i-3j**), further supporting our conclusion that Th1 differentiation was less sensitive to the loss of CD25 than Th2 differentiation.

2. *GATA3 is very poorly resolved in all cases. Basing conclusions on this weak signal is probably ill advised.*

We apologize for the lack of clarity in GATA-3 staining in our data. We agree with the Reviewer that the staining of GATA-3 with the anti-GATA3 antibody (clone 16E10A23) is typically modest even with overnight staining in our hands, but it is important to note that the comparison with the ndLN always shows a clear shift in the dLN (**Fig. 2j**). To clarify, in addition to % of GATA-3+ cells in OT-II, we have now added MFI of GATA-3 in total OT-II cells (**Fig.2k, Supplementary Fig.S2e and S3m**).

3. *The statement “these results indicate that CD301b+ DCs promote Th2 cell differentiation through reinforcing the availability of IL-2 for cognate CD4 T cells by the DC-intrinsic expression of CD25” does not make sense to this reviewer.*

We apologize for any lack of clarity in our original manuscript. In this study, we show that CD301b⁺ DC-intrinsic CD25 is required for the Th2 differentiation as well as for STAT5 phosphorylation in OT-II cells but not in endogenous Treg cells (**Fig. 4r-4t**), which indicates that CD301b⁺ DC-intrinsic CD25 promotes the IL-2R signaling selectively in the cognate CD4T cells that they are actively interacting with. Since CD25 in CD301b⁺ DCs does not seem to efficiently signal in CD301b⁺ DCs themselves (see our response to point #5 by Reviewer #1 “Possible regulation of CD301b⁺ DC numbers by autocrine IL-2”), we speculate that CD301b⁺ DCs use their own CD25 to trans-present IL-2 selectively to the cognate CD4T cells, preventing it from diffusing and being taken up by the surrounding Treg cells. While this manuscript is focused on the functional consequences of such interaction rather than providing concrete evidence for trans-presentation of IL-2, we have now included a schematic figure describing this model in **Fig 8**.

4. *It was not obvious to this reviewer that the scRNA sequencing results add much to the story. The various CD40 ablation or blockade experiments are much stronger evidence that “CD301b+ DC-intrinsic CD40 is required for Th2 cell differentiation”.*

We agree with the Reviewer that the functional studies shown in **Fig.6** alone would be sufficient to claim the functional requirement of CD301b⁺ DC-intrinsic CD40 in Th2 cell fate instruction. However, we nevertheless believe that the CITEseq data would strengthen our conclusion and are also of interest to the readers, since it shows differential expression of the CD40 pathway – not simply just CD40 alone – in CD301b^{hi} cDC2s, which also highlights functional specialization of CD301b^{hi} cDC2s compared not only to LCs and cDC1s but also to CD301b^{lo} cDC2s. We wish to keep the CITEseq data in **Fig.5** as is, since the data shown in **Fig.6** alone do not reveal subset-specific phenotypic differences to this granularity.

5. *The Tfh marker CXCR5 is barely detected in Fig. 7A. The PD-1hi PSGL-1lo phenotype is not a standard way to identify Tfh cells. Absent evidence that these cells disappear in a Bcl-6-deficient animal*

it is not clear that Tfh cells are being detected. In my opinion, the paper would be stronger without this section.

The rationale for looking into the bifurcation between Tfh and non-Tfh cells in this study is that we previously reported unusual expansion of CXCR5⁺Tfh cells in CD301b⁺DC-depleted mice 7 days after immunization with OVA plus papain (*Elife* 5:e17979). While we agree that PD-1^{hi}PSGL-1^{lo} phenotype is not a standard way to identify fully mature Tfh cells, it has been shown that the downregulation of PSGL-1 in Tfh cells is dependent on Bcl6 using *Bcl6*^{-/-} OT-II cells and *Bcl6*^{-/-} mice (*J Immunol* 185:313). In fact, the upregulation of Bcl6 and downregulation of PSGL-1 precedes and does not require T–B cell interaction, which is instead required for the subsequent upregulation of CXCR5 (*J Immunol* 185:313). Since CD301b⁺DCs are the first antigen presenting cell population for naïve antigen-specific CD4T cells to interact with upon dLN entry (*Sci Immunol* 6:eabg0336), we thus focused on the very early event in the Tfh – non-Tfh bifurcation that presumably happens *prior* to the upregulation of CXCR5. To further clarify the Tfh identify of the PD-1^{hi}PSGL-1^{lo} cells, we have now added data showing increased Bcl6 expression in OT-II and endogenous CD4 T cells primed with OVA plus papain in the CD301b^{AIIL-2} mice (**Fig 7m-n**), further confirming the higher frequencies of pre-Tfh cells in the absence of CD301b⁺DC-derived IL-2. While it has been established that IL-2 regulates the Tfh – non-Tfh bifurcation, the activated T cells themselves are generally believed to be the main source of IL-2 in the current model (*Science* 361:eaao2933). The data in **Fig. 7** show the involvement of DC-derived IL-2 in this process for the first time as far as we know, and therefore is a part of the fundamental message of this manuscript. Please also see our response to the Minor Point # 8 by the Reviewer #3.

Reviewer #3 (Remarks to the Author)

The manuscript of Tatsumi et al seeks to identify the dendritic cell-derived signals that program Th2 activation and differentiation. This is an important but still unanswered question and this manuscript offers original and interesting information on this topic.

The authors use an adoptive OT-II transfer model together with targeted genetic modification of CD301b⁺ DC in skin lymph node to dissect the early interaction between OT-II cells and DCs, focusing on the role of IL-2, CD25 and CD40 signaling in this process. They conclude that CD40-dependent IL-2 production by CD301b⁺ DC, and their expression of CD25, are necessary for Th2 differentiation but not for CD4⁺ T cell proliferation or Th1 differentiation. While a role for IL-2 signaling in Th2 differentiation has been reported (eg ref 77 in manuscript) and is well accepted, this manuscript proposes a completely new mechanism by showing that DCs, and specifically CD301b⁺ DCs, are an important source of such IL-2.

The manuscript uses many sophisticated conditional KO models, and carefully timed characterizations of the lymph node immune response. While some of the conclusions are well supported by experimental evidence, others were less clear or not as convincingly demonstrated.

We appreciate the Reviewer #3 for positive feedback and constructive criticism. We have now added more data and edited the writing to address the concerns. Please see our responses to each comment below.

Major points:

1) The gating of CD301b⁺ and CD301b⁻ populations should follow standard DC gating strategies, e.g. separating DC into migratory vs LN resident, and then into DC1, DC2 and LC as in PMID: 20038600. This gating strategy should then be kept consistent throughout the study. Instead, gating of CD301b⁺ and CD301b⁻ populations is inconsistent across experiments, and especially so for the CD301b⁻ population which appears to include LC, DC1 and resident DC2 populations in 4B, but only DC2s in S3H. Without a rigorous DC gating, evaluating the impact of the conditional deletions becomes impossible.

We apologize for the lack of clarity in the original manuscript. We have now revised gating strategies for DC subsets to distinguish between migratory DCs (MHCII^{hi}CD11c⁺) and LN-resident DCs (MHCII^{int}CD11c^{hi}) throughout the study and also have separated LCs, cDC1s and cDC2s in Fig.5h, 5j, and **Supplementary Fig.S4m-S4o** and **S5h-S5m**) as suggested. In addition, to further clarify the specificity of Cre expression in the Mgl2^{+Cre} mice, we have now added reporter expression data from the Mgl2^{+Cre};Rosa26^{LSL-ITom} mice in the new **Supplementary Fig.S1**, showing that the expression of Mgl2-Cre is largely restricted to CD301b⁺ PD-L2⁺ migratory (MHCII^{hi}) cDC2s and a part of CD301b⁻ PD-L2⁺ migratory cDC2s (please also see our response to the point #5 by the Reviewer #1, "the number of CD301b⁺ DCs in the Mgl2^{+Cre} mice") but is largely absent from other cell types including the CD301b⁻ PD-L2⁻ migratory cDC2s and LN-resident cDC2s, cDC1s and LCs.

As to the separation of CD301b⁻ DC subsets, as the Reviewer pointed out, there are some inconsistencies in gating strategy between different experiments as different combinations of markers were used in different experiments due to limitations in filters on our cytometers. For instance, we were not able to have LC-specific markers in Fig.4b, Fig.5i and **Supplementary Fig.S4a** and **S4b** (former Fig.4B and 4C), thus the CD301b⁻ CD11b⁺ migratory DCs in these data include both LCs and CD301b⁻ migratory cDC2s (the labeling has been revised). Likewise, we were not able to fit XCR1 in our staining panel in **Supplementary Fig.S4c-S4e** and instead used CD103 as a cDC1 marker, and were thus unable to separate the LN-resident DCs into cDC1 and cDC2. However, we nevertheless believe that our data are fully interpretable for "evaluating the

impact of the conditional deletions", because (1) the expression of Cre in Mgl2^{+/Cre} mice is largely specific to CD301b⁺ migratory cDC2s (**Supplementary Fig.S1**), and (2) the conditional deletion of target genes in CD301b^{ΔIL-2} (**Supplementary Fig.S4e**), CD301b^{ΔCD25} (**Supplementary Fig.S4n**), and CD301b^{ΔMHCII} (Fig.S2F in *Sci Immunol* 6:eabg0336) mice was indeed specific to CD301b⁺ migratory cDC2s and not observed in LCs, cDC1s or in LN-resident DCs.

2) Could the authors please clarify whether the CD301b⁺ DC2s represent a separate subset within the migDC2s lineage, or simply a maturation stage (with all migDC2s expressing the CD301b marker at some stage of their maturation), providing references if available. This is important for understanding the results. I could not find this information in previous publications from the group.

We thank the Reviewer for this insightful question. While we agree with the importance of understanding the differentiation mechanism of CD301b⁺ DCs and we are currently investigating into it, we would like to focus on their function and not their development in this manuscript. At this moment, we do not fully understand the developmental relation of CD301b⁺ DCs to other cDC2 subsets, but we nevertheless believe that CD301b⁺ DCs represent a *functionally* distinct subset of cDC2s that is different from CD301b⁻ cDC2s for the following reasons:

First, we previously reported that CD301b⁺ DCs suppress Tfh differentiation and class-switching of B cells (*Elife* 5:e17979), which is also supported by the data shown in **Fig.7** of this manuscript. In contrast, others have shown that mice lacking IRF4 specifically in CD11c⁺ cells fail to develop Tfh cells due to their developmental and/or functional defect in cDC2 cells (*Immunity* 39:722; *Sci Immunol* 2:eaam9169). Since the *CD11c-Cre;Irf4^{fl/fl}* mice lack both CD301b⁻ and CD301b⁺ cDC2 cells in the skin-dLN while the Mgl2-DTR mice only lack the latter (*Immunity* 39:722; *Immunity* 39:733), this suggests that cDC2s contain at least two functionally distinct subsets – one that suppresses Tfh and is positive for CD301b and one that promotes Tfh and is presumably negative for CD301b.

Second, as shown in **Supplementary Fig.S1** and discussed in the Point #1 above, the expression of Mgl2-Cre is largely restricted to CD301b⁺ migratory (MHCII^{hi}) cDC2s. Importantly, however, the expression levels of MHCII and CD86 are comparable between Mgl2-Cre⁺ and Mgl2-Cre⁻ migratory cDC2s in the Mgl2-Cre mice (**Supplementary Fig.S1c, S1d**), or between CD301b⁺ and CD301b⁻ migratory cDC2s in WT mice (**Supplementary Fig. S1e-S1g**), suggesting comparable maturation levels between these two subsets. Likewise, we observed specific deletion of MHCII, IL-2 and CD25 selectively in CD301b⁺ DCs in CD301b^{ΔMHCII}, CD301b^{ΔIL-2} and CD301b^{ΔCD25} mice, respectively (Fig.S2E and S2F in *Sci Immunol* 6:eabg0336, **Supplementary Fig.S4c-S4e, Fig.S4m** and **S4n** in this manuscript). Thus, although we cannot directly address the Reviewer's speculation (that is, *all migDC2s expressing the CD301b marker at some stage of their maturation*) at this moment, we believe that it would affect neither the interpretation nor the novelty of this study, as the experimental models we use do not seem to affect the phenotype or development of CD301b⁻ cDC2s.

3) The authors state (in title and conclusions) that the CD40-IL-2-CD25 axis is not involved in Th1 priming. However, the IFN γ responses to OVA-papain (which is used throughout the study) is only about 1-2% of OT-II, compared to 8-10 % of OT-II expressing IL-4. While FCA gave better IFN γ responses in some expts (Fig3V), this was not always the case (eg Fig S1C). Therefore, the effect on Th1 could not be evaluated because little to no IFN γ was made after immunization. A robust protocol of Th1 induction is needed.

We apologize for the lack of clarity in our previous Th1 data. As suggested, we have now added new data from a strongly Th1-biased response induced by immunization with OVA plus CpG (**Fig.1i-1k, Supplementary Fig. S2a-S2c, Fig.3d-3j**). Please also see our responses to the point #1 by the Reviewer #2 ("Role of CD301b⁺ DCs in priming and Th1 differentiation of OT-II cells upon immunization with OVA plus CpG" and "Role of IL-2R signaling in CD4 T cells for Th1 differentiation").

4) *IL-2 expression in CD301b⁺ DCs is a key step of the proposed mechanism of Th2 priming. However, IL-2 expression in DCs is shown only after PMA+Iono stimulation, and is very low in the CD301b⁺ subset. The authors need to provide evidence of IL-2 protein expression by DC in vivo, or alternatively by anti-CD40-stimulated DCs ex vivo. It would be interesting to see if the Il2 gene can be resolved in the scRNAseq dataset shown in figure 5.*

We apologize for the lack of clarity in the intracellular IL-2 staining in our original data. We have now revised the gating strategy and showed IL-2 staining for each DC subset (**Fig.4b and Supplementary Fig. S4a, S4b**). While we agree with this Reviewer that the IL-2 expression in CD301b⁺DCs is indeed low compared to OT-II cells, but in fact it is comparable to endogenous T cells (**Fig.4b and Supplementary Fig. S4b**). In addition, the staining of intracellular IL-2 in CD301b⁺ DCs is not necessarily low (7.86% in CD301b⁺ DCs and 9.54% in host T cells, **Supplementary Fig. S4b**) and can be clearly distinguished from the isotype control staining (for example, **Supplementary Fig. S4d**). As suggested by this Reviewer, we have also now analyzed the CITEseq data for the *Il2* gene expression in CD301b⁺ DCs and other DCs freshly isolated from the dLNs of papain-immunized mice (**Fig.5d**). Consistent with previous reports (ref 45-49), our data show that multiple DC subsets have a capacity to produce IL-2 in our model.

That said, it is indeed technically challenging to obtain a clear staining of IL-2 without *ex vivo* stimulation with PMA and ionomycin, and we did not observe clear IL-2 staining upon *ex vivo* stimulation of CD40 either, thus there seems to be an essential cue for IL-2 production that is not fully recapitulated by simply stimulating CD40 *in vitro*. However, it is important to note that *ex vivo* restimulation with PMA and ionomycin was also necessary for detecting intracellular IL-2 even in the OT-II cells in our hands, so we believe that the poor staining of IL-2 in cells without *ex vivo* restimulation does not necessarily mean low IL-2 production level by those cells. Please also see our response to the Point # 5 by the Reviewer #1 ("IL-2 expression in CD301b⁺DCs").

5) *The manuscript uses many elegant models of conditional gene KO in CD301b⁺ DCs and antibody-mediated blocking, but does not state whether these mice were assessed to determine potential impacts of mutation/treatments on eg T cell differentiation and homeostasis, numbers and ratios of conventional T cells and Tregs, etc. For example, increased IL-2 production by DCs has been associated with increased Treg frequency (Whyte et al PMID: 35699942). Was Treg frequency altered in the CD301b-DIL-2?*

We appreciate the Reviewer for this question. The OT-II cells do not express any detectable Foxp3 in our model even in WT and WT-equivalent recipients (for instance, "control" in **Supplementary Fig.S4g, S4h**). Some of the endogenous polyclonal CD4T cells express Foxp3 as expected, but the frequency and number of both FoxP3⁺ Tregs and Foxp3⁻ Tconvs were comparable between the control and CD301b^{All-2} mice (**Supplementary Fig.S4g, S4h**), indicating that CD301b⁺ DC-derived IL-2 is dispensable for the overall maintenance of Tregs. This is consistent with the report by Owen et al (*J Immunol* 200:3926) that deletion of IL-2 specifically in CD11c⁺ cells does not affect Treg homeostasis in peripheral lymphoid organs. Likewise, we previously showed that the depletion of CD301b⁺ DCs does not affect Foxp3 expression in the total CD4T cells and in Tfr cells in the skin-dLNs (Fig.8d in *Elife* 5:e17979).

While the study mentioned by the Reviewer (Whyte *et al*, *J Exp Med* 219:e20212391) shows that overexpression of IL-2 specifically but systemically in cDCs increases Tregs in the spleen, there are very few, if any, CD301b⁺ DCs in the spleen (Fig.2f in *Elife* 5:e17979) and thus the impact of overexpression of IL-2 in CD301b⁺ DCs remains unclear. Of note, a recent study by Weckel *et al*. shows the requirement of CD301b⁺ DCs for the development of Tregs against skin commensals in neonates (*Immunity* 56:1239), but the role of CD301b⁺ DC-derived IL-2 in that process also remains unknown.

6) *The introduction presents a model where the differentiation of various Th subsets (Th1, Th2...) is exclusively determined by CD4+ T cell priming in the context of different DC subsets. This does not account for other publications showing promiscuity, which is consistent with, for example, DC subsets other than DC1 being able to produce IL-12 when exposed to different stimuli (eg, PMID: 33159073 11466361). Broadening the introduction to include other models would in my view better reflect existing literature as well as provide a better starting point to approach the results.*

We appreciate the Reviewer for this insightful suggestion. We have now added the following sentences citing the two papers mentioned by the Reviewer in the first paragraph of the Introduction: “the capacity to induce specific types of Th cells is not always confined to the same DC subset, as DC subsets other than cDC1s can be a critical source of IL-12 for inducing Th1 cell differentiation when exposed to certain stimuli.”

7) *The manuscript Figures are large and crowded (for example, figures 3 and 4 each include more than 30 panels) and could be made easier to follow and see by moving all non-essential data to Supplemental.*

We have now moved some data in the original Fig. 3 and Fig. 4 to **Supplemental Fig. S3** and **Fig. S4**, respectively.

Minor:

1) *An important role of T-cell intrinsic CD25 expression in Th1 and Th2 effector differentiation was demonstrated in PMID: 22018468 and 26750312, these papers should be cited.*

We now have cited PMID: 26750312 and 22018468 in ref 33 and 89 in the revised version.

2) *Can the authors rule out pSTAT5 (Figs 2, 3, 4) being induced during sample preparation rather than in vivo? The LN digestion step (30' at 37C according to the Methods) is long enough to extinguish IL-2 signaling that had taken place in vivo.*

For detecting pSTAT5, LNs are fixed immediately upon harvest without enzymatic digestion, ruling out the possibility that pSTAT5 was induced during sample preparation. This was now clarified in the Methods section.

3) *Several Figures show frequencies of OT-II cells expressing specific markers, but often there are no clear positive and negative populations to be identified (Figs 2J-K, 3I...). Comparing median fluorescence intensity of the whole population would be more appropriate in these cases.*

We now have included the median fluorescence intensity of GATA-3 in total OT-II cells as shown in **Fig.2k** and **Supplementary. Fig.S3m** for GATA-3 expression. Please also see our response to the point #2 by the Reviewer #2.

4) Figure 3T-U shows very different frequencies of IL-2ra^{+/+} vs. IL-2ra^{-/-} T cells in the same mouse after immunization (3T), but identical numbers of divisions/no division in the two populations (3U). How can these data be reconciled? Please discuss.

We do not fully understand the exact reason for the reduction of IL-2ra^{-/-} OT-II cells compared to the IL-2ra^{+/+} counterpart, as there were no significant differences in CFSE dilution or in cell death (Fig.3g, 3h). We speculate that the reduction was due to the poor survival of the primed IL-2ra^{-/-} OT-II cells immediately after the priming (i.e., before the first cell division) and/or the difference in cell divisions after the CFSE gets fully diluted (i.e., the number of cells that have divided more than 8-9 times). We have now added the following sentence to the main text: "The reduced expansion was not due to poor survival (Fig. 3h), suggesting that it was instead due to differences in proliferation that was not detectable within the range of CFSE dilution we used."

5) Figure 3V-W and S1F: I agree that there might be stronger inhibition in the IL-4⁺ populations, nonetheless the IFN γ populations are also clearly inhibited and this should be acknowledged. Differences in inhibition might be simply due to differences in the dose response of each cytokine.

Please see our response to the major point #3 above.

6) What is the difference between the FCA experiments in 3V-W and S1G-I? The legend to 3V-W does not mention FCA but FCA is included in 3R-W.

We now have revised Fig.3 and the legend.

7) 6P: here the response is expressed as % divided cells in the CD44⁺ population. Comparing % divided in the total OT-II population would enable a more quantitative comparison.

Fig.6p indeed shows % CD44⁺ divided (CFSE^{lo}) cells out of the total OT-II cells, not % divided cells out of the CD44⁺ cells. We have edited the Y-axis titles to clarify this.

8) There is no CXCR5 staining in Figure 7. Either the staining did not work, or the time of analysis was not appropriate.

In Fig. 7a, the CXCR5 staining is actually working in the endogenous CD4 T cells (1.45 \pm 0.22%) on day 7 in the control mice immunized with OVA and papain, compared to the isotype staining control (0.04%). Albeit low, this CXCR5 expression level is typical to this model (for instance, see "WT" in Fig.8A in *Elife* 5:e17979). Since CXCR5 is only faintly detectable in OT-II cells in the same LN, we instead focused on the immature Tfh cells that have upregulated Bcl6 and downregulated PSGL1 but not yet expressed CXCR5 (*J Immunol* 185:313) (Fig. 7k-7n). Please also see our response to the point # 5 by the Reviewer #2.

Reviewer's Comments:

Our responses are shown in blue below:

Reviewer #1 (Remarks to the Author):

The authors revised the manuscript in accordance with reviewers' comments, which increased its clarity and impact.

We thank this reviewer for constructive comments and appreciate their time and effort in reviewing our manuscript.

Reviewer #3 (Remarks to the Author):

I would like to thank the authors for updating their manuscript and making several points clearer including DC gating and phenotype, a discussion of CD301b⁺ as a subset vs maturation marker, background on the cKO mouse strains used etc.

We thank this reviewer's critical but positive feedback on our manuscript and appreciate their time and effort in reviewing our manuscript.

However, my major point 3 has been addressed only in part. While Figures 1 and 3 have been modified to include a model of OVA+CpG immunization which gives strong Th1 priming, and nicely confirm the Authors' conclusions, other figures including Figs 3, 4 and 6 still rely on very low IFN γ measurements after immunization in Th2 conditions using papain+OVA to conclude that findings in Th2 setting do not extend to Th1 settings. This conclusion cannot be based on OVA+papain data and require data from an OVA+CpG experiment. Papain+OVA does not elicit a Th1 response thus the relative panels are not informative and should be removed together with the related comments and discussion.

Similarly, in Figure 3j, the authors compare normalized IL-4 and IFN γ responses to each other, to conclude that "Th2 cell differentiation relies more stringently on IL-2R signaling than Th1". However, any differences in 3j are only due to low responses in 3i. Panel 3j should be removed from the Figure together with the "more stringent" claims which in my opinion are not justified.

We apologize for any lack of clarity in the manuscript and rebuttal. The original Major point #3 by this Reviewer in the previous review reads as follows: *The authors state (in title and conclusions) that the CD40-IL-2-CD25 axis is not involved in Th1 priming. However, the IFN γ responses to OVA-papain (which is used throughout the study) is only about 1-2% of OT-II, compared to 8-10 % of OT-II expressing IL-4. While FCA gave better IFN γ responses in some expts (Fig3V), this was not always the case (eg Fig S1C). Therefore, the effect on Th1 could not be evaluated because little to no IFN γ was made after immunization. A robust protocol of Th1 induction is needed.*

To clarify, it is not our intent to dismiss the role of CD40, IL-2 or CD25 in Th1 cell priming in general. As we state in the Discussion that "the present study reveals that the DC-intrinsic CD40-IL-2 axis in CD301b⁺ DCs plays a crucial role in Th2 cell fate instruction by quantitatively regulating the IL-2R signaling in CD4T cells", our study is entirely focused on the role of CD301b⁺ DC-intrinsic molecules rather than the general role of CD40, IL-2 or CD25, which we believe is accurately reflected in the Title, Abstract, and elsewhere in this manuscript. Further, our data in Figure 1 clearly show that CD301b⁺ DC-dependent cognate interaction is *dispensable* for Th1 differentiation induced by papain, alum, or CpG, which is also consistent with our previous observation of intact Th1 differentiation in CD301b⁺ DC-depleted mice (*Immunity* 2013; *Sci Immunol* 2021). We therefore focused on the role of CD301b⁺ DC-intrinsic CD40, IL-2 and CD25 *only* in the context of Th2-polarizing (papain) condition in Figures 4 and 6.

However, strictly in the context of CD301b⁺ DC-intrinsic mechanism and in the papain-induced Th cell differentiation, we did draw a conclusion that the CD40-IL-2-CD25 axis in CD301b⁺ DCs is *not* involved in Th1 priming in Figure 4. For example, we state "the differentiation of Th2 cells...was abolished in the CD301b^{ΔIL-2} recipients without affecting...their differentiation into

IFN γ -producing Th1 cells (Fig. 4g-k)", which we believe is an accurate description of our data. It is, however, important to note that we did not draw any conclusion about Th1 priming under Th1-polarizing conditions in the main text for Figure 4, because we have not tested these conditions such as OVA plus CpG.

Likewise, in Figure 6, we did not draw any conclusion regarding Th1 priming under Th1-polarizing conditions. For instance, we believe that our statement "the OT-II cells primed with OVA plus papain in the CD301b^{ACD40} BMC mice had significantly reduced Th2 cells but not Th1 cells (Fig. 6g)" is perfectly accurate under the experimental conditions used for this Figure, but we did not draw any conclusion regarding Th1-polarizing conditions because it was not the focus of these experiments. We nevertheless believe that showing *the lack of change* in Th1 differentiation in the Th2-polarizing condition in these Figures is actually informative, because the manipulation of CD301b⁺ DCs can potentially result in a decrease of Th1 differentiation if T cell activation *per se* was impaired, or, alternatively, it can result in an *increase* of Th1 differentiation if Th2 cells were induced *at the cost of* Th1 fate decision (neither of which was the case according to our data). Therefore, we wish to keep the IFN γ data in Figures 4 and 6 as is.

In Figure 3, we examined the role of CD4T cell-intrinsic CD25 in Th cell differentiation and concluded that "the Th2 fate decision by antigen-specific CD4 T cells *in vivo* requires a potent and cell-intrinsic IL-2R signaling and is more sensitive to a partial loss of CD25 than the Th1 cell differentiation." Unlike Figures 4 and 6 in which we focused only on the Th2-polarizing condition, Figure 3 is one of the few cases (other than Figure 1) where we actually compared Th1- and Th2-polarizing conditions because we examined a CD4T cell-intrinsic mechanism rather than a DC-intrinsic mechanism in this Figure.

This Reviewer argues that Figure 3 (as well as Figures 4 and 6) "*rely on very low IFN γ measurements after immunization in Th2 conditions using papain+OVA to conclude that findings in Th2 setting do not extend to Th1 settings*" and that "*this conclusion cannot be based on OVA+papain data and require data from an OVA+CpG experiment*". However, we would like to point out that we did not attempt to simply "extend" our findings on Th1 differentiation under the Th2 condition (papain) to Th1 differentiation under other conditions in this Figure. Instead, we actually examined all conditions and compared Th2 and Th1 differentiation in each and across conditions.

In fact, while this Reviewer argues that "*Panel 3j should be removed from the Figure*", we believe that the analysis shown in Figure 3j is critical for comparing between conditions and provided us with the following observations, which was only possible by normalizing the data to the WT OT-II cells transferred in the same recipients:

(1) While papain induces more Th2 cells than FCA, the Th2 cells were reduced to a similar level in *Il2ra*^{+/-} OT-II cells in both models when normalized to their WT counterpart in the same host (21.6 \pm 4.2% for papain [p<0.0001 compared to WT OT-II] and 10.4 \pm 3.9% for FCA [p<0.0001 compared to WT OT-II], with no statistically significant difference [p=0.0673] when comparing the *Il2ra*^{+/-} OT-II cells between papain and FCA models), indicating that the partial loss of CD25 affected Th2 differentiation similarly between the papain- and FCA-immunized mice. Th2 cells were not detectable when the mice were immunized with CpG, but similar reduction of Th2 cells was observed for the *Il2ra*^{+/-} OT-II cells between the papain- and FCA-immunized hosts (4.7 \pm 1.4% for papain [p=0.0005 compared to WT OT-II] and 17.1 \pm 3.6% for FCA [p<0.0001 compared to WT OT-II], with a modestly significant difference [p=0.0184] when comparing *Il2ra*^{+/-} OT-II cells between papain and FCA). Thus, the impact of the loss of CD25 on Th2 cells was somewhat similar between the papain- and FCA-immunized hosts.

(2) Th1 cell differentiation was most efficiently induced with CpG, followed by FCA and minimally induced with papain (yet higher than naïve, as we showed previously in Figure 7B in *Immunity* 39:733). Despite this variability in Th1 priming efficacy between different models, when normalized to the WT OT-II cells in the same host, the Th1 differentiation was reduced to a similar level in *Il2ra*^{+/-} OT-II cells (78.1 \pm 20.2% for papain [p=0.3188 compared to WT OT-II], 63.6 \pm 14.0% for FCA [p=0.0043 compared to WT OT-II], and 61.6 \pm 11.1% for CpG [p=0.0003

compared to WT OT-II]), though the reduction in the papain-immunized mice did not reach statistical significance. Importantly, there was no statistically significant difference in the WT-normalized Th1 differentiation in *Il2ra*^{+/-} OT-II cells between different models. Likewise, the Th1 differentiation in *Il2ra*^{-/-} OT-II cells was significantly reduced compared to the WT counterpart across different models (40.4±9.9% for papain [p=0.0046 compared to WT OT-II], 43.0±7.8% for FCA [p<0.0001 compared to WT OT-II], and 38.6±7.4% for CpG [p=0.0024 compared to WT OT-II]), whereas no statistically significant difference was observed in the WT-normalized Th1 differentiation in *Il2ra*^{-/-} OT-II cells between different models. Thus, despite the wide variability in Th1 differentiation efficacy in different models, just like Th2 differentiation, these data indicate surprisingly similar impact of the loss of CD25 on Th1 differentiation.

(3) The comparison between WT-normalized Th1 and Th2 differentiation consistently shows a significantly more impact of the loss of CD25 on the latter. When normalized to the WT counterpart, the reduction of Th2 differentiation is consistently greater than the reduction of Th1 differentiation within the same model. For example, in the *Il2ra*^{+/-} OT-II cells immunized with papain, the Th2 differentiation was reduced to 21.6±4.2% while Th1 was reduced to 78.1±20.2% (p=0.011). The Th2 differentiation of FCA-immunized *Il2ra*^{+/-} OT-II cells was reduced to 10.4±3.9% while Th1 was reduced to 63.6±14.0% (p=0.0018). More importantly, these differences remain consistent when comparing different models. For instance, the Th2 differentiation in both papain- and FCA-immunized mice was reduced more significantly than the Th1 differentiation in CpG-immunized mice (p=0.0061 vs. papain-induced Th2; p=0.0005 vs. FCA-induced Th2), which remains the same for WT-normalized Th1 and Th2 differentiation in the *Il2ra*^{-/-} OT-II cells (p=0.0051 for papain-Th2 vs. papain-Th1; p=0.0084 for FCA-Th2 vs. FCA-Th1; p<0.0001 for papain-Th2 vs. CpG-Th1). Thus, our conclusion that “the Th2 fate decision by antigen-specific CD4 T cells *in vivo* requires a potent and cell-intrinsic IL-2R signaling and is more sensitive to a partial loss of CD25 than the Th1 cell differentiation” remains true for all comparisons we tested.

(4) While we agree with this Reviewer that papain may not be the best model to evaluate Th1 differentiation, the above observations indicate that, when normalized to the WT OT-II cells in the same host, the impact of the loss of CD25 *does not* correlate with the magnitude of the Th1 or Th2 differentiation and is constantly greater for Th2 differentiation than for Th1 differentiation. In fact, the magnitude of Th1 differentiation of WT OT-II cells in mice immunized with OVA plus FCA (5.75±0.67%) was similar to the magnitude of Th2 differentiation of WT OT-II cells in mice immunized with papain (8.71±0.84%) and was greater than the Th2 differentiation induced by FCA (3.51±0.37%) (Figure 3i), but the loss of CD25 suppressed the FCA-induced Th2 differentiation more significantly (reduced to 10.4±3.9% compared to WT OT-II) than the FCA-induced Th1 differentiation (63.6±14.0%). Thus, this Reviewer’s interpretation “*any differences in 3j are only due to low responses in 3i*” does not appear to be the case, since there is no correlation between the magnitude of response and the magnitude of suppression by the loss of CD25. We therefore believe that our data collectively support our conclusion and wish to keep the data as is.

To clarify these points, we have now added p-values and mean±SEM to Figures 3i and 3j and clarified the writing.

Minor:

Fig S1A, it is surprising that all XCR1⁺ cells are also CD326⁺

In fact, it is well known that cDC1 cells (which are XCR1⁺) in the skin and lung in mice universally express CD326 (EpCAM) along with other epithelial cell and Langerhans cell markers such as CD103 and CD207, though its expression level is lower than that in Langerhans cells. For instance, it is reported in Figure 4 in *J Exp Med* 204:3147, which is one of the first studies that identified CD207⁺ dermal DCs as a subset (currently known as cDC1) that is developmentally unrelated to Langerhans cells.

The statement “reduced expansion was not due to poor survival (Fig. 3h), suggesting that it was instead due to differences in proliferation that was not detectable within the range of CFSE dilution we used” does not make sense.

We apologize for the lack of clarity in this statement. As mentioned in our response to this Reviewer’s original Minor point #4 in the previews review, we do not fully understand the exact reason for the reduction of *Il2ra*^{-/-} OT-II cells compared to the *Il2ra*^{+/+} counterpart, as we did not see any difference in survival or cell cycle progression (i.e., CFSE dilution) between WT and *Il2ra*^{-/-} OT-II cells. While the reason for the reduced expansion of *Il2ra*^{-/-} OT-II cells is not the main focus of this study and therefore we have no experimental evidence, we now speculate that it was due to impaired survival and/or recruitment of naïve OT-II cells to the dLN (i.e, less naïve *Il2ra*^{-/-} OT-II cells entered the dLN compared to WT OT-II cells due to poor survival in circulation and/or impaired homing). The main text was revised accordingly.

Data in 7A show that there are almost no TFH cells in all conditions (or the staining didn’t work), this panel should be removed.

We apologize for the lack of clarity in this Figure. As we mentioned in our response to the original Minor point #8 by this Reviewer, the low expression level of CXCR5 and the relatively low percentage of Tfh cells (both out of OT-II and endogenous CD4T) are a typical phenotype of this model. However, the gating for Tfh was set based on the isotype control staining for CXCR5 (Figure 7a) as well as on the staining in the non-draining LN (not shown in this manuscript, but see Figure 8e in *Elife* 5:e17979 for example) and thus we are confident that our CXCR5 staining was working properly in all of our experiments. We have now refined our gating and the plotting format in Figure 7a to clarify this point, and also revised the main text accordingly.

As to the low Tfh differentiation and the lack of difference in % Tfh cells between the control and CD301b^{ΔMHCII} mice in Figure 7a, as clearly mentioned in the main text (“Unlike CD301b⁺ DC-depleted Mgl2-DTR mice¹⁹”), what we are trying to emphasize in this particular Figure is the difference from the CD301b⁺ DC-depleted mice we previously reported (*Elife* 5:e17979), in which we observed a substantial increase in fully mature CXCR5⁺PD-1⁺ Tfh cells in both OT-II and endogenous CD4T cell compartments compared to CD301b⁺ DC-intact mice. While our subsequent data (Figure 7c-7n) show more significant increases in the PSGL1^{lo} PD-1⁺ Bcl6⁺ “immature” Tfh cells (see ref.69 for their definition), we did not observe a clear increase in mature CXCR5⁺PD-1⁺ Tfh in Figure 7a, thus we concluded that “more CD4T cells are poised for Tfh differentiation in these mice” instead of saying that they are “committed” for Tfh. Since *the lack of robust mature Tfh differentiation in the CD301b^{ΔMHCII} mice (in contrast to the clear increase in Mgl2-DTR mice shown in ref.19) is an important component to draw this conclusion, we believe that the data in Figure 7a is informative for readers and wish to keep the data as is (with the revised gating).*

Reviewer #4 (Remarks to the Author):

The authors have adequately addressed this reviewers concerns.

We thank this reviewer for constructive comments and appreciate their time and effort in reviewing our manuscript.

Minor point: It is not stated which statistical test was used to determine the significance of the multiple comparisons in Fig, 4b.

We apologize for this oversight. It is now added to the Legend.